# `DocPruner`: A Storage-Efficient Framework for Multi-Vector Visual Document Retrieval via Adaptive Patch-Level Embedding Pruning

## Abstract

Visual Document Retrieval (VDR), the task of retrieving visually-rich document pages using queries that combine visual and textual cues, is crucial for numerous real-world applications. Recent state-of-the-art methods leverage Large Vision-Language Models (LVLMs) in a multi-vector paradigm, representing each document as patch-level embeddings to capture fine-grained details. While highly effective, this approach introduces a critical challenge: *prohibitive storage overhead*, as storing hundreds of vectors per page makes large-scale deployment costly and impractical. To address this, we introduce `DocPruner`, **the first framework to employ adaptive patch-level embedding pruning for VDR** to effectively reduce the storage overhead. `DocPruner` leverages the intra-document patch attention distribution to dynamically identify and discard redundant embeddings for each document. This adaptive mechanism enables a significant 50-60% reduction in storage for leading multi-vector VDR models with negligible degradation in document retrieval performance. Extensive experiments across more than ten representative datasets validate that `DocPruner` offers a robust, flexible, and effective solution for building storage-efficient, large-scale VDR systems.

## 1 Introduction

Visual Document Retrieval (VDR), the task of retrieving relevant document pages based on a query that leverages both visual and textual cues, is of paramount importance in numerous real-world applications, from e-commerce product searches to educational resource discovery (Ding et al., 2024; Zheng et al., 2025; Wang et al., 2025b). In contrast to traditional text retrieval, VDR presents a greater challenge as it must interpret not only the textual content but also the complex layouts, tables, figures, and other visual elements that convey critical information (Abootorabi et al., 2025; Mei et al., 2025). Consequently, this intricate task has garnered increasing attention within the information retrieval community in recent years, driving innovation beyond text-centric paradigms.

The methodology for VDR has undergone a significant paradigm shift. Early approaches were predominantly **OCR-based**, involving the extraction of text from document images, which was then indexed by conventional text retrievers (Zhang et al., 2024a; Hegghammer, 2022), as shown in Figure 1 (a). However, these methods are often brittle and error-prone, frequently *failing to preserve the vital layout and structural relationship inherent in the visual representation* (Most et al., 2025; Guo et al., 2025). With the recent advent of Large Vision-Language Models (LVLMs) and their dramatically enhanced visual understanding capabilities (Caffagni et al., 2024), the research community has begun to explore **LVLM-based** methods, which have demonstrated state-of-the-art retrieval performance (Macé et al., 2025; Günther et al., 2025; Dong et al., 2025; Tanaka et al., 2025). These methods generally fall into two categories: one that encodes an entire document page and the query into single, holistic embeddings (*i.e.,* page-level retrieval) (Zhang et al., 2024c; Liu et al., 2025b; Jiang et al., 2024b; Meng et al., 2025), and another that represents a document as multiple patch-level embeddings and the query as multiple token-level embeddings, as illustrated in Figure 1 (b). The former approach, while simple, often *fails to capture the fine-grained details* necessary for understanding complex documents, leading to suboptimal performance. As a result, the latter patch-level retrieval has emerged as the preferred paradigm for leading models.

**Figure 1:** The illustration of comparison between OCR-based (a) & LVLM-based (b) paradigms for VDR, and `DocPruner` (c), a novel framework to adaptively prune the patch-level embeddings for diverse document types.

The ascent of the patch-level retrieval paradigm is primarily attributed to the advantages of multi-vector retrieval, a technique pioneered by ColBERT-style late interaction (Khattab & Zaharia, 2020). The core mechanism of this approach involves a `MaxSim` operation, where for each query token embedding, the maximum similarity score against all patch embeddings of a document is computed, and these scores are then aggregated to determine relevance. The VDR field first witnessed the successful application of this paradigm with ColPali (Faysse et al., 2024), which spurred a wave of subsequent works that further refined and enhanced the performance of multi-vector VDR (Nom-icAI, 2025; Günther et al., 2025; Xu et al., 2025a; Karlinsky et al., 2025). However, despite its effectiveness, *the multi-vector approach suffers from a critical efficiency bottleneck:* ***prohibitive storage overhead***. Storing hundreds or even thousands of embedding vectors for every single document page makes large-scale deployment costly and challenging (Ma et al., 2025).

To address this critical challenge, we introduce `DocPruner`, the **first framework to successfully employ adaptive pruning in the context of VDR** to significantly alleviate storage overhead, as shown in Figure 1 (c). The core of `DocPruner` is an elegant yet powerful mechanism that leverages the patch-level attention score distribution within a single document to perform adaptive pruning of its patch embeddings. This allows the framework to dynamically adjust the pruning ratio for different documents, achieving a 50-60% reduction in patch embeddings for several state-of-the-art multi-vector models with negligible performance degradation. While some prior works have explored efficiency optimizations for multi-vector VDR, they are often constrained by pre-defined pruning rates or fixed thresholds (Cheng et al., 2024a; Ma et al., 2025; Tmamna et al., 2024), which lack the adaptability required for diverse, real-world visual documents. We believe that the design philosophy of `DocPruner`, which enables robust performance even for diverse models and datasets, ensures its flexibility and extensibility for practical, large-scale multimodal retrieval applications.

Our contributions can be summarized as follows:

❶ ***Pioneering Pruning for VDR.*** We propose `DocPruner`, the first framework to introduce an adaptive pruning mechanism to the VDR domain. It achieves a substantial 50-60% average patch pruning rate with near-lossless performance, effectively mitigating the storage overhead of top-performing multi-vector VDR models.

❷ ***Adaptive Property for Diverse Documents.*** The adaptive nature of `DocPruner` allows it to dynamically tailor the pruning ratio for different types of visual documents, a feature that is particularly crucial in real-world scenarios where document formats and information densities vary widely.

❸ ***Extensive Experimental Validation.*** We conduct comprehensive experiments on diverse and even multilingual VDR benchmarks, demonstrating the effectiveness and robustness of `DocPruner` when integrated with multiple leading multi-vector retrieval models in the community.

## 2 RELATED WORK

### 2.1 VISUAL DOCUMENT RETRIEVAL

Visual Document Retrieval (VDR) aims to retrieve relevant visually-rich documents based on visual representations, a paradigm that has garnered significant attention from the research community (Zheng et al., 2025; Mei et al., 2025; Zhang et al., 2025b). Previous **OCR-based methods** rely on document parsing to extract textual content (Xiao et al., 2024; Wang et al., 2023; Karpukhin et al., 2020), a process that can *lose critical layout information and fail to interpret non-textual components*. Consequently, the field has rapidly evolved from early OCR-plus-retriever pipelines to

a paradigm leveraging powerful VLMs as OCR-free retriever backbones. By treating documents as images, VDR systems can preserve this vital structural and visual integrity, enabling a "what-you-see-is-what-you-get" retrieval mechanism that aligns with human perception (Ma et al., 2024).

These **VLM-based methods** primarily fall into two categories: efficient but less detailed page-level retrieval, and the more powerful patch-level retrieval. **Page-level retrievers**, such as DSE (Ma et al., 2024), GME (Zhang et al., 2024c) and UniSE (Liu et al., 2025b), encode an entire document page into a single, compact embedding. While efficient, this approach may *lose fine-grained details crucial for specific queries*. State-of-the-art **patch-level retrievers** (*e.g.,* ColPali (Faysse et al., 2024), ColQwen (Faysse et al., 2024), ColNomic (NomicAI, 2025), Jina Embeddings v4 (Günther et al., 2025), and Llama Nemoretriever Colembed (Xu et al., 2025a)) achieve superior performance by generating fine-grained, multi-vector representations per page, yet this introduces a critical bottleneck due to *prohibitive storage and computational overhead*. Our proposed `DocPruner` directly addresses this pain point by proposing a solution to adaptively reduce the storage footprint of patch-level embeddings, thereby making high-performance VDR more practical and scalable.

## 2.2 MULTI-VECTOR RETRIEVAL

Multi-vector retrievers, also known as late-interaction models (Khattab & Zaharia, 2020; Ji et al., 2024), computes relevance by first independently encoding queries and documents into sets of token-level embeddings and then performing fine-grained similarity calculations. Formally, given a query $q$ and a document $d$ with $L_1$ and $L_2$ tokens respectively, they are encoded into embedding matrices $\mathbf{Q} = (\mathbf{q}_1, \ldots, \mathbf{q}_{L_1}) \in \mathbb{R}^{P \times L_1}$ and $\mathbf{D} = (\mathbf{d}_1, \ldots, \mathbf{d}_{L_2}) \in \mathbb{R}^{P \times L_2}$, where $P$ is the embedding dimension. The final score is derived from their token-wise similarity matrix $\mathbf{S} = \mathbf{Q}^\top \mathbf{D}$. For instance, ColBERT model (Khattab & Zaharia, 2020) computes the score via a MaxSim operation:

$$s(q, d) = \sum_{i=1}^{L_1} \max_{j=1}^{L_2} \mathbf{q}_i^\top \mathbf{d}_j. \tag{1}$$

Building on this foundation, ColBERTv2 (Santhanam et al., 2021) introduced a centroid-based method to compress token embeddings for greater storage efficiency. PLAID (Santhanam et al., 2022) further optimized this by using centroid interactions for efficient pruning of low-scoring documents. Other approaches have focused on reducing the number of stored vectors: XTR (Lee et al., 2023) trains the model to prioritize and retrieve only key document tokens, Acquavia et al. (2023) remove embeddings of less impactful tokens, and Clavié et al. (2024) cluster similar token embeddings at indexing time to reduce the total vector count. Recently, MUVERA (Jayaram et al., 2024) proposed using Fixed Dimensional Encodings (FDEs) to approximate the multi-vector similarities, enabling efficient retrieval. Despite their effectiveness, a primary limitation of these text-based multi-vector models is their *significant storage overhead*, which scales linearly with the number of document tokens ($L_2$), resulting in a storage cost of $O(P \times L_2)$ per document, a substantial increase compared to $O(P)$ cost of single-vector models (MacAvaney et al., 2025; Ji et al., 2024).

The concept of multi-vector retrieval has been extended to VDR, leveraging the fine-grained interaction capabilities to better align textual queries with visual content (Plale et al., 2025; Xu et al., 2025b). Pioneering this direction, ColPali (Faysse et al., 2024) adapted the ColBERT framework by using PaliGemma-3B model (Beyer et al., 2024) to generate multi-vector embeddings directly from document images. Subsequently, Llama Nemoretriever Colembed (Xu et al., 2025a) further advanced this paradigm by modifying Llama-3.2-3B (Grattafiori et al., 2024) with bidirectional attention and employing a two-stage training strategy to achieve state-of-the-art performance on the ViDoRe benchmark. More recently, Jina Embeddings v4 (Günther et al., 2025) proposed a unified Qwen2.5-VL (Bai et al., 2025) architecture that supports both single-vector and multi-vector outputs, utilizing LoRA adapters for task-specific optimization. However, the storage overhead problem persists in these visual models, which remains a critical challenge that `DocPruner` aims to address.

More related work can be seen in Appendix A.

## 3 METHODOLOGY

In this section, we first formalize the setting of multi-vector VDR in (▷ Section 3.1). We then introduce our proposed framework, `DocPruner`, detailing its mechanism for adaptive patch-level

embedding pruning in ($\triangleright$ Section 3.2). Finally, we establish a theoretical foundation rooted in information theory to justify its efficacy in ($\triangleright$ Section 3.3).

## 3.1 TASK FORMULATION

The task of VDR is to retrieve a ranked list of relevant document pages from a large corpus $\mathcal{C} = \{d_1, d_2, \ldots, d_{|\mathcal{C}|}\}$ for a given textual query $q$. In the context of multi-vector VDR (Faysse et al., 2024), both queries and documents are represented by sets of embedding vectors.

Let a query $q$ be a sequence of $L_q$ textual tokens. A VLM-based encoder, denoted as $\Phi(\cdot)$, maps this query into a set of token-level embeddings $\mathbf{Q} = \{\mathbf{q}_i\}_{i=1}^{L_q}$, where each $\mathbf{q}_i \in \mathbb{R}^P$ and $P$ is the embedding dimension. Similarly, a document page $d$ is first rendered as an image and then processed by the VLM encoder $\Phi(\cdot)$, which divides the image into a grid of patches. This process yields a set of $L_d$ patch-level embeddings $\mathbf{D} = \{\mathbf{d}_j\}_{j=1}^{L_d}$, where each $\mathbf{d}_j \in \mathbb{R}^P$.

Following the late-interaction paradigm (Khattab & Zaharia, 2020; Santhanam et al., 2021), the relevance score $s(q, d)$ is computed via a `MaxSim` operation as defined in Equation 1. The primary challenge is the storage overhead associated with this representation. Storing the full set of embeddings $\mathbf{D}$ for every document results in a cost of $O(L_d \times P)$ per page, which is prohibitive for large-scale corpora. Our objective is to generate a pruned set of document embeddings $\mathbf{D}' \subset \mathbf{D}$ such that its size, $L_d' = |\mathbf{D}'|$, is significantly smaller than $L_d$ ($L_d' \ll L_d$), thereby substantially reducing the storage cost to $O(L_d' \times P)$ while preserving retrieval performance.

## 3.2 THE `DocPruner` FRAMEWORK

`DocPruner` is a lightweight, plug-and-play framework applied during the offline indexing phase. It is designed around two core principles: being **query-agnostic** to enable offline processing and **document-adaptive** to handle the diverse nature of visual documents. The framework systematically identifies and discards redundant or less informative patch embeddings without requiring any model retraining. The process involves three main steps: quantifying patch importance, applying an adaptive threshold, and scoring with the pruned embeddings. See pseudocode in Section B.

### 3.2.1 QUANTIFYING PATCH IMPORTANCE VIA GLOBAL TOKEN ATTENTION

The central challenge of offline pruning is to *estimate the importance of each patch without access to a query*. We need a reliable, intrinsic signal of salience. Our key insight is that a VLM, in the process of understanding a document image, already computes such a signal. Specifically, we leverage the attention mechanism directed towards a **global token**. A global token is a special token whose final hidden state is trained to aggregate and summarize information from the entire input sequence. Its representation must encapsulate the document's overall semantics.

In our framework, we use the end-of-sequence `[EOS]` token as the default global token, a common and effective choice in many VLM architectures. We extract the attention weights from the final Transformer layer, as this layer captures the most abstract and semantically rich relationships.

Formally, let $\mathbf{A}^{(L)}$ be the attention weights from the final layer $L$. After averaging across all $H$ attention heads to create a smooth, robust attention map ($\bar{\mathbf{A}}_{i,j}^{(L)} = \frac{1}{H} \sum_{h=1}^{H} \mathbf{A}_{h,i,j}^{(L)}$), we define the importance score $I(\mathbf{d}_j)$ for the $j$-th patch as the attention it receives from the global token:

$$I(\mathbf{d}_j) = \bar{\mathbf{A}}_{\text{global},j}^{(L)}. \tag{2}$$

This process yields a vector of importance scores $\mathcal{I}_d = \{I(\mathbf{d}_j)\}_{j=1}^{L_d}$ for each document, which serves as the foundation for our adaptive pruning.

### 3.2.2 ADAPTIVE THRESHOLDING FOR PRUNING

Naive pruning strategies, such as using a fixed pruning ratio or a global threshold, are ill-suited for VDR. Visual documents exhibit vast heterogeneity in information density—a sparse title page has very different characteristics from a dense, text-filled page. A fixed strategy would either over-prune the dense page, losing critical information, or under-prune the sparse page, retaining useless

background patches. `DocPruner`'s adaptive thresholding directly addresses this by tailoring the pruning decision to the statistical properties of each individual document.

For a given document $d$ with $L_d$ patch embeddings, we have a corresponding vector of importance scores $\mathcal{I}_d = \{I(\mathbf{d}_j)\}_{j=1}^{L_d}$. Our method computes a document-specific threshold by leveraging the first two statistical moments of these scores. First, we define the **mean importance** $\mu_d$, which establishes a baseline salience level for the document's patches. A high mean suggests the document is generally information-rich. It is formally calculated as:

$$\mu_d = \frac{1}{L_d} \sum_{j=1}^{L_d} I(\mathbf{d}_j). \tag{3}$$

Second, we compute the **standard deviation** $\sigma_d$, which measures the dispersion of importance scores. A high standard deviation indicates that a few patches are exceptionally important compared to the rest, a hallmark of sparse but salient content. It is calculated as:

$$\sigma_d = \sqrt{\frac{1}{L_d} \sum_{j=1}^{L_d} (I(\mathbf{d}_j) - \mu_d)^2}. \tag{4}$$

The adaptive pruning threshold $\tau_d$ for document $d$ is then defined as a linear combination of these two statistics: $\tau_d = \mu_d + k \cdot \sigma_d$, where $k$ is a hyperparameter that acts as a adaptation factor. It determines how many standard deviations above the mean a patch's importance score must be considered significant. We define the preliminary pruned set of patch embeddings $\hat{\mathbf{D}}'_d$ as:

$$\hat{\mathbf{D}}'_d = \{\mathbf{d}_j \in \mathbf{D}_d \mid I(\mathbf{d}_j) > \tau_d\}. \tag{5}$$

To handle the edge case where overly aggressive pruning might discard all embeddings (i.e., $\hat{\mathbf{D}}'_d = \emptyset$), we guarantee that at least one embedding is preserved. The final pruned set $\mathbf{D}'_d$ is defined as:

$$\mathbf{D}'_d = \begin{cases} \hat{\mathbf{D}}'_d & \text{if } \hat{\mathbf{D}}'_d \neq \emptyset \\ \{\mathbf{d}_{j^*}\} \text{ where } j^* = \underset{j \in \{1, \dots, L_d\}}{\arg\max} I(\mathbf{d}_j) & \text{if } \hat{\mathbf{D}}'_d = \emptyset. \end{cases} \tag{6}$$

### 3.2.3 SCORING WITH PRUNED EMBEDDINGS

The ultimate goal of pruning is to reduce storage and, by extension, accelerate online retrieval, without compromising ranking quality. At query time, the retrieval process remains identical to the standard late-interaction paradigm, with one crucial difference: the search space for the `MaxSim` operation is significantly reduced. Instead of comparing each query token embedding against the full set of document embeddings $\mathbf{D}$, we use the compact, pruned set $\mathbf{D}'$. The pruned relevance score, $s'(q, d)$, is computed as: $s'(q, d) = \sum_{i=1}^{L_q} \max_{\mathbf{d}_j \in \mathbf{D}'} \mathbf{q}_i^\top \mathbf{d}_j$. For a given query $q$, we compute $s'(q, d_k)$ for all documents $d_k$ in the corpus to obtain a ranked list. The effectiveness of this ranking is then evaluated using Normalized Discounted Cumulative Gain at rank 5 (nDCG@5).

### 3.3 THEORETICAL FOUNDATION

The efficacy of `DocPruner` can be rigorously analyzed through the **Information Bottleneck (IB) principle** (Tishby et al., 2000; Saxe et al., 2019; Tishby & Zaslavsky, 2015). The IB framework aims to learn a compressed representation $\mathbf{Z}$ of an input random variable $\mathbf{X}$ that is maximally informative about a target variable $\mathbf{Y}$. This is formulated as the following optimization problem:

$$\max_{\mathbf{Z}} \quad \mathcal{L}_{IB}(\mathbf{Z}) = I(\mathbf{Z}; \mathbf{Y}) - \beta I(\mathbf{Z}; \mathbf{X}), \tag{7}$$

where $I(\cdot; \cdot)$ denotes mutual information and $\beta$ is a Lagrangian multiplier balancing compression and information preservation.

**The Intractable Ideal.** In our VDR task, $\mathbf{X}$ is the full set of document embeddings $\mathbf{D}$, $\mathbf{Z}$ is the pruned set $\mathbf{D}'$, and the target $\mathbf{Y}$ is the relevance score $s(q, d)$, which depends on a future, unknown query $q$. The ideal objective is to maximize the expected information about relevance over the distribution of all possible queries $P(q)$:

$$\max_{\mathbf{D}'} \quad \mathbb{E}_{q \sim P(q)}[I(\mathbf{D}'; s(q, d))] \quad \text{s.t.} \quad |\mathbf{D}'| \ll |\mathbf{D}|. \tag{8}$$

This objective is intractable due to the unknown query distribution $P(q)$.

`DocPruner` **as a Tractable Approximation.** `DocPruner` offers a principled, tractable approximation to this problem.

▶ **Global Token as Relevance Proxy**. The hidden state of global token, $\mathbf{h}_{global}$, serves as a sufficient statistic for document's relevance to an arbitrary query. That is, $I(\mathbf{D}; s(q, d)) \approx I(\mathbf{D}; \mathbf{h}_{global})$. This axiom posits that the global token's representation, which summarizes the entire document, captures the necessary information for determining relevance. The attention scores $I(\mathbf{d}_j)$ directly measure the information flow from each patch to this summary. Therefore, by selecting patches that maximize $I(\mathbf{D}'; \mathbf{h}_{global})$, we are effectively approximating the ideal, intractable objective.

▶ **Entropy-Aware Pruning.** The adaptive threshold $\tau_d$ dynamically adjusts the pruning ratio based on the information entropy of the document's attention distribution. Let the normalized attention scores form a probability distribution $p_d(j) = \frac{I(\mathbf{d}_j)}{\sum_i I(\mathbf{d}_i)}$ over the patches. The information content of the document is captured by its Shannon entropy $H(p_d) = -\sum_j p_d(j) \log p_d(j)$.

1. **Low-Entropy Documents:** For documents with low information entropy (*e.g.,* title pages), $p_d$ is a sparse, peaky distribution. A few patches have very high attention scores, while most have near-zero scores. The term $k \cdot \sigma_d$ dominates, setting a high threshold $\tau_d$ that isolates only the highly informative "outlier" patches, resulting in aggressive pruning.

2. **High-Entropy Documents:** For documents with high information entropy (*e.g.,* dense text pages), $p_d$ is more uniform. Attention scores are distributed more evenly across many patches. The threshold $\tau_d$ is more lenient, preserving a larger patch number that collectively contribute to the document's meaning.

## 4 EXPERIMENT

### 4.1 EXPERIMENTAL SETUP

**Benchmarks & Evaluation.** We conduct our experiments on recent representative VDR benchmarks: **ViDoRe-V2** (Macé et al., 2025) and **JinaVDR-Bench** (Günther et al., 2025) (More details in Appendix C). We use three state-of-the-art multi-vector VDR models as our base models: **ColQwen2.5** (Faysse et al., 2024), **ColNomic** (NomicAI, 2025), and **Jina Embeddings V4** (Günther et al., 2025). Following standard practice in VDR domain (Faysse et al., 2024; Günther et al., 2025; NomicAI, 2025; Xu et al., 2025a), we use **nDCG@5** as the primary evaluation metric.

**Baselines.** We compare `DocPruner` against three categories of baselines.

**(I) Base Models.** This represents the original multi-vector models without any pruning or merging. They serve as the performance upper bound of storage cost.

**(II) Merging-based Methods.** Following Ma et al. (2025), the only work focused on VDR storage optimization via merging, we implement three merging strategies:

▶ **Sem-Cluster:** Merges patch embeddings by performing hierarchical clustering and representing each cluster by its centroid. The tunable hyperparameter is the `merging_factor`, which determines the target number of clusters.

▶ **1D-Pooling:** Applies 1D average pooling over sequential groups of patch embeddings to reduce their count. The hyperparameter is `merging_factor`, which defines the pooling window size.

▶ **2D-Pooling:** Arranges patch embeddings into a 2D grid and applies 2D average pooling. The hyperparameter is `merging_factor`, which must be a perfect square.

**(III) Pruning-based Methods:** We compare three pruning strategies adapted to VDR context:

▶ **Random:** Randomly discards a fixed fraction of patch embeddings, serving as a naive baseline. The hyperparameter is the `pruning_ratio`.

▶ **Attention-plus-Similarity:** An adaptive method that computes a combined score from both the `[EOS]` attention (importance) and the embedding similarity to the `[EOS]` token (representativeness), then prunes patches below a dynamically calculated threshold, following Wen et al. (2025). Hyperparameters include an adaptive factor `k` and a weighting factor `alpha`.

▶ **Pivot-Threshold:** A two-stage adaptive method that first filters an "important set" of patches using an adaptive attention threshold, and then de-duplicates this set by pruning patches that are too similar to selected "pivot", following VisPruner (Zhang et al., 2025c). Hyperparameters include an adaptive factor for importance `k`, a de-duplication factor `k_dup`, and `num_pivots`.

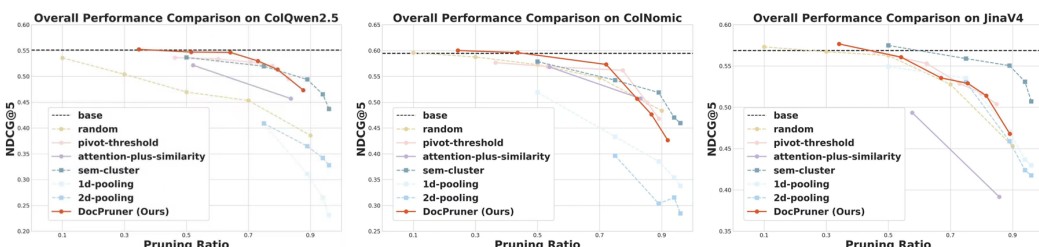

**Figure 2:** Performance comparison (nDCG@5) between `DocPruner` and baselines on ViDoRe-V2 benchmark (Macé et al., 2025) across ColQwen2.5 (*Left*), ColNomic (*Middle*), and Jina Embedding V4 (*Right*). Here, *solid lines* denote adaptive methods, whereas *dashed lines* denote non-adaptive ones; *circular nodes* represent pruning methods, whereas *square nodes* represent merging methods.

**Implementation Details.** To ensure fair and reproducible comparisons, we replicated the base results of three base models in aligned with their respective official implementation. Our evaluation codebase is adapted from the official ViDoRe Benchmark repository[1]. The complete code for our experiments, including all baseline implementations and the `DocPruner` framework, will be made publicly available upon acceptance. For `DocPruner`, the adaptation factor $k$ has a range of $\{-0.5, -0.25, 0, 0.25, 0.5, 1\}$. The details of hyperparameters for all baseline methods is detailed in the Appendix D. All experiments were conducted on a NVIDIA A100 (80GB) GPU cluster.

### 4.2 EXPERIMENTAL ANALYSIS

In this section, we conduct a comprehensive experimental analysis to answer four key research questions (RQs). (**RQ1**) How effectively does `DocPruner` maintain retrieval performance on diverse visual document types while achieving significant storage compression? (**RQ2**) Can `DocPruner`'s robust performance generalize to multilingual retrieval scenarios? (**RQ3**) What is the difference between `DocPruner` framework and its variants? (**RQ4**) What are the quantifiable relative improvements in storage efficiency and latency of implementing `DocPruner`?

#### 4.2.1 RETRIEVAL PERFORMANCE COMPARISON (RQ1)

To answer RQ1, we evaluate `DocPruner`'s performance against a comprehensive set of baselines on the ViDoRe-V2 benchmark. The results, visualized in Figure 2, demonstrate the effectiveness and robustness of our approach across three leading multi-vector models. See more results in Sec.E.1.

**Observation ❶: `DocPruner` achieves near-lossless retrieval performance while pruning 50-60% of embeddings, demonstrating remarkable robustness across different base models.** As illustrated in Figure 2, `DocPruner` consistently operates near the performance ceiling set by the unpruned base models (*i.e.,* dashed black line) even when aournd 60% of embeddings are pruned. For instance, when applied to ColQwen2.5, `DocPruner` removes 51.6% of patch embeddings with a mere 0.0038 drop in nDCG@5 (from 0.5508 to 0.5470). This high efficiency is mirrored on Jina Embedding V4, where it prunes 54.1% of embeddings while the nDCG@5 only decreases from 0.5687 to 0.5608. Even on the high-performing ColNomic model, `DocPruner` achieves a 43.6% pruning ratio with a negligible performance change (0.5960 vs. the base's 0.5946), showcasing a remarkable balance between efficiency and accuracy. This robustness stems from `DocPruner`'s mechanism, which leverages intra-document attention to create a document-specific importance score for each patch, effectively retaining the most semantically salient information necessary for retrieval.

**Observation ❷: Pruning-based strategies are generally more effective at preserving retrieval performance than merging-based strategies.** This trend is evident across all three models, where methods marked with circles (pruning) consistently form a higher-performance frontier than those with squares (merging). For instance, on the ColNomic model at a around 75% compression ratio, `DocPruner` achieves an nDCG@5 of 0.5730 (at a 72.1% ratio), whereas the strongest merging baseline, `sem-cluster`, drops to 0.5426 (at a 75% ratio). The reason for this disparity is that merging, by averaging feature vectors, can dilute the distinctiveness of highly salient patches, blurring important signals. In contrast, pruning preserves the original, high-fidelity embeddings of the most critical patches, vital for the late-interaction mechanism's ability to find precise query-patch matches.

---

[1] https://github.com/illuin-tech/vidore-benchmark

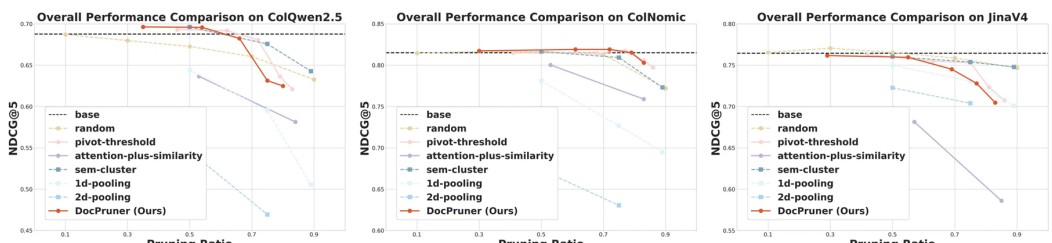

**Figure 4:** Performance comparison (nDCG@5) between `DocPruner` and baselines on JinaVDR benchmark (Günther et al., 2025) across ColQwen2.5 (*Left*), ColNomic (*Middle*), and Jina Embedding V4 (*Right*). Here, *solid lines* denote adaptive methods, whereas *dashed lines* denote non-adaptive ones; *circular nodes* represent pruning methods, whereas *square nodes* represent merging methods.

**Observation ❸: Adaptive pruning methods generally exhibit a superior performance-compression trade-off compared to non-adaptive, fixed-ratio approaches.** The solid lines in Figure 2, representing adaptive methods like `DocPruner`, consistently maintain higher nDCG@5 scores than their non-adaptive counterparts (dashed lines) at similar compression levels (*esp.,* below 60% ratio). For example, on the ColNomic

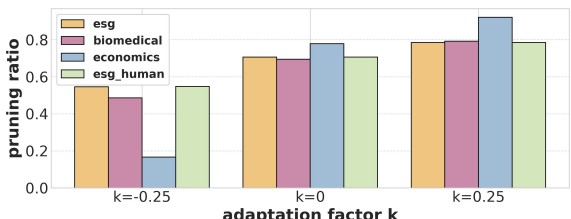

**Figure 3:** Adaptive pruning ratio values of four different datasets in ViDoRe-V2 across difference k (More in Sec.E.1).

model, `DocPruner` achieves a high nDCG@5 of 0.5960 with a 43.6% pruning ratio, outperforming all non-adaptive baselines. The superiority of adaptive methods is because they intelligently account for the heterogeneity of visual documents (validated by Figure 3); they prune more aggressively on information-sparse pages and more conservatively on information-dense ones, whereas fixed-ratio methods apply a one-size-fits-all strategy that can be suboptimal.

**Observation ❹: Notably, merging-based methods exhibit uncharacteristically strong performance on the Jina Embedding V4, in some cases surpassing `DocPruner`.** This phenomenon can likely be attributed to JinaV4's unique training architecture; its technical report (Günther et al., 2025) reveals that the model is explicitly co-trained to produce a single-vector embedding via `mean_pooling` over its token-level representations. This training paradigm encourages the model to learn patch embeddings that are inherently more aggregable and robust to averaging, making post-hoc merging strategies unusually effective as they align with the model's intrinsic properties.

### 4.2.2 GENERALIZATION TO MULTILINGUAL SCENARIOS (RQ2)

To answer RQ2, we evaluate `DocPruner`'s generalization capability on the multilingual JinaVDR benchmark, where we choose documents in German, Russian, Chinese, and Japanese. The overall and per-language results, presented in Figure 4 and Sec.E.2, lead to the following observations.

**Observation ❺: `DocPruner` demonstrates strong and consistent performance across diverse multilingual datasets, maintaining near-lossless retrieval accuracy while achieving substantial storage savings (*i.e.,* around 50-60%).** For instance, on the ColNomic model, `DocPruner` achieves a remarkable 61.0% overall pruning ratio with a slight increase in nDCG@5 from the base's 0.8151 to 0.8191. Similarly, when applied to ColQwen2.5, it prunes 54.0% of embeddings while improving the nDCG@5 score from 0.6877 to 0.6958. This robust generalization stems from `DocPruner`'s core mechanism, which relies on the *language-agnostic* visual attention patterns within the VLM.

**Observation ❻: `DocPruner`'s adaptive nature is particularly evident in its ability to dynamically adjust pruning ratios for documents in different languages, reflecting varying information densities.** This tailored approach is clearly visible in the per-language pruning statistics shown in Appendix E.2. Using the ColNomic model as an example (with k=-0.5), `DocPruner` applies a modest pruning ratio of 9.0% for German documents (nDCG@5 of 0.6022 vs. base 0.5975) and 7.0% for Spanish documents (nDCG@5 of 0.7896 vs. base 0.7927). In contrast, it identifies greater redundancy in other languages, pruning 36.3% for Japanese and 37.6% for Chinese documents while maintaining high performance. This demonstrates that `DocPruner` is not applying a uniform rule

but is sensitive to the intrinsic properties of the documents themselves, automatically allocating the storage budget proportional to each document's information entropy.

### 4.2.3 VARIANT STUDY (RQ3)

To answer RQ3, we conduct a variant study comparing `DocPruner` against pruning-based variants (shown in Figure 5), which are: **(I) attention-ratio**, a non-adaptive method that prunes a fixed percentage of patches with the lowest attention scores; **(II) attention-threshold**, which uses a fixed, global attention value as the pruning threshold; and **(III) attention-threshold-nfp**, which enhances the static threshold method with a noise-filtering-prompt (nfp) to guide the model's focus.

**Observation ❼:** **The document-adaptive statistical thresholding of `DocPruner` consistently achieves a superior performance-compression trade-off compared to simpler pruning variants that rely on fixed ratios or static thresholds.** While all methods leverage attention scores, their pruning criteria differ fundamentally: attention-ratio enforces a uniform compression rate, whereas attention-threshold and attention-threshold-nfp apply a one-size-fits-all importance cutoff. At a significant pruning ratio of approximately 60%, `DocPruner` sustains a high nDCG@5 of 0.54; but the performance of the static attention-threshold variant collapses to below 0.45, and even the improved attention-threshold-nfp and fixed-ratio attention-ratio methods lag considerably.

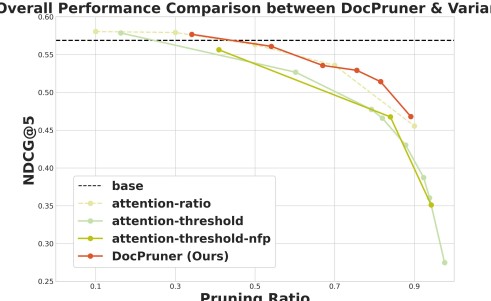

**Figure 5:** Overall comparison between `DocPruner` & variants (See per-dataset analysis in Sec.E.3).

### 4.2.4 EFFICIENCY ANALYSIS (RQ4)

**Observation ❽:** `DocPruner` **achieves a substantial storage footprint reduction of approximately 50% on average with near-lossless retrieval performance, at the cost of an acceptable increase in offline encoding latency.** As detailed in Table 1, `DocPruner` reduces storage footprints by 51.55% for ColQwen, 43.62% for ColNomic, and 54.09% for JinaV4, while the nDCG@5 performance changes are minimal (↓0.69%, ↑0.24%, and

**Table 1:** Relative improvement of **performance**, **storage**, and **latency** to base models on ViDoRe-V2 (adaptation factor k as -0.25; orange denotes better and green denotes worse).

| △ | ColQwen | ColNomic | JinaV4 |
|---|---|---|---|
| **nDCG@5** | ↓0.69% | ↑0.24% | ↓1.39% |
| **Storage** | ↓51.55% | ↓43.62% | ↓54.09% |
| **Latency** | ↑60.00% | ↑65.96% | ↑66.00% |

↓1.39%, respectively). This specific setting of k=-0.25 consistently delivers an optimal trade-off between performance and storage across all multi-vector models. Although `DocPruner` introduces an overhead that increases offline latency by 60-66% due to the extra steps of attention score extraction and filtering, the practical impact is modest. The average per-document encoding time increases from a baseline of 0.47s to only 0.77s, a duration that is acceptable for an offline indexing phase and vastly superior to 7.22s required by OCR-based method (*i.e.,* OCR+BGE-M3 (Chen et al., 2024a)).

## 5 CONCLUSION

In this paper, we addressed the critical challenge of prohibitive storage overhead in state-of-the-art multi-vector VDR systems. We introduced `DocPruner`, a novel and adaptive framework for patch-level embedding pruning, which leverages the attention paid by a global token to each image patch to derive a query-agnostic importance score. Crucially, `DocPruner` employs a document-specific statistical threshold, allowing it to dynamically adjust the pruning ratio for documents of varying information density and complexity. Through extensive experiments across more than ten benchmark datasets, we have demonstrated that `DocPruner` can achieve a substantial 50-60% reduction in stored patch embeddings with only negligible degradation in retrieval accuracy. Future work could explore integrating this pruning mechanism directly into the model training process or extending the adaptive principle to other modalities. Ultimately, `DocPruner` charts a path toward fine-grained multimodal understanding as practical, real-world applications at an unprecedented scale.

We elaborate the broader impact of `DocPruner` and LLM usage in Section F and G, respectively.

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

## Technical Appendices and Supplements

## A   MORE RELATED WORK

### A.1   LARGE VISION-LANGUAGE MODELS

Large Vision-Language Models (LVLMs) have recently revolutionized a multitude of fields, including visual question answering (Borisova et al., 2025; Zakari et al., 2022; Jang et al., 2025), urban sensing (Zou et al., 2025; Yan et al., 2024c; Yan & Lee, 2024; Hou et al., 2025b), multimodal reasoning (Wang et al., 2024; Yan et al., 2025a; 2024a; Su et al., 2025b; Yan et al., 2024b), multimodal retrieval (Lin et al., 2024; Lu & Tang, 2025; Kagaya et al., 2024; Zhong et al., 2024), and visual document understanding (Li et al., 2024c; Zhang et al., 2025a; Ding et al., 2024; Hu et al., 2024a;b). The architecture of these models generally follows several key paradigms. The first involves connecting a pre-trained vision encoder (*e.g.,* ViT) and a LLM via a lightweight projection module, as seen in models like BLIP-2 (Li et al., 2023). A second paradigm consists of end-to-end trained models that process visual and textual inputs within a unified architecture, such as PaliGemma (Beyer et al., 2024). A third, highly effective approach involves freezing the core vision and language backbones and fine-tuning lightweight adapters (*e.g.,* LoRA) to bridge the modalities, a strategy popularized by LLaVA (Li et al., 2024b;a). Furthermore, recent research is actively optimizing these models for critical real-world requirements, such as minimizing hallucination (Bai et al., 2024; Zhou et al., 2024; Zheng et al., 2024; Zhu et al., 2024), enabling agent-based interaction (Xie et al., 2024; Yan et al., 2025b; Su et al., 2025a; Durante et al., 2024), and enhancing interpretability (Lin et al., 2025; Huo et al., 2024; 2025; Huang et al., 2025) and safety (Fang et al., 2025; Chen et al., 2025; Liu et al., 2025a). The multi-vector models evaluated in our work are built upon such powerful LVLMs; for instance, ColQwen (Faysse et al., 2024) and ColNomic (NomicAI, 2025) are based on the Qwen2.5-VL series (Bai et al., 2025), one of the leading open-source LVLMs, while Jina Embeddings v4 (Günther et al., 2025) further leverages this foundation to implement a unified training paradigm for both single-vector and multi-vector outputs.

### A.2   PRUNING IN LVLMS

The extensive length of visual token sequences in LVLMs poses significant computational challenges, motivating a surge of research in token compression (Cheng et al., 2024a; Tmamna et al., 2024; Ye et al., 2025a). These training-free methods primarily fall into two paradigms. The first is **instruction-centric pruning** (Hou et al., 2025a; Huang et al., 2024; Federici et al., 2024), which leverages query-document interaction. Methods like FastV (Chen et al., 2024b) and Sparse-VLM (Zhang et al., 2024d) identify redundant visual tokens by analyzing the cross-attention scores between textual instructions and visual patches. While effective for tasks like VQA, this paradigm is fundamentally *incompatible with the offline indexing phase of VDR*, as it requires a query to determine token importance. The second paradigm is **vision-centric compression** (Ye et al., 2025b; Jiang et al., 2024a), which is query-agnostic and thus more suitable for offline processing. This category includes token merging approaches like ToMe (Bolya et al., 2022), which progressively combines similar tokens, and token pruning methods like FasterVLM (Zhang et al., 2024b), which uses the attention scores of the `[CLS]` token within the vision encoder to rank and discard less salient patches. However, these vision-centric methods often suffer from their own limitations, such as information dilution from merging or retaining redundant tokens due to the concentrated nature of attention. Crucially, most pruning strategies are designed for and evaluated on generative tasks, and their direct application to the offline retrieval setting is underexplored (Lassance et al., 2023; Acquavia et al., 2023; Liu et al., 2024). They are *not tailored to preserve the fine-grained, discriminative features essential for the late-interaction mechanism in multi-vector retrieval*.

### A.3   EFFICIENT DOCUMENT RETRIEVAL

The pursuit of efficiency in multi-vector retrieval (Wu et al., 2024; Park et al., 2025; Shrestha et al., 2024; Bian et al., 2025; Scheerer et al., 2025), a challenge amplified in the visual domain, has been addressed through two main orthogonal approaches: **Dimension Reduction** and **Token Reduction**. Dimension reduction aims to shrink the size of each embedding vector (Su et al., 2021; Yoon et al., 2024; Wang et al., 2025a). A prominent example is ColBERTv2 (Santhanam et al., 2021),

which employed product quantization to compress embeddings. This principle was later inherited by ColPali (Faysse et al., 2024), which uses a simpler projection layer for the same purpose. The second, more impactful approach is token reduction, which focuses on decreasing the number of vectors stored per document and can be divided into pruning and merging strategies (Liu et al., 2023; Mao et al., 2025; Cheng et al., 2024b). However, recent empirical studies (Ma et al., 2025) have highlighted that token merging strategies, which aggregate multiple embeddings into a smaller set of representative vectors (*e.g.,* via spatial pooling or semantic clustering (Clavié et al., 2024)), are considered more appropriate for the offline VDR context as they retain information from all patches. Our work, `DocPruner`, revisits the pruning paradigm by introducing a novel *adaptive, query-agnostic mechanism that sidesteps the pitfalls of static pruning*, offering a storage-efficient alternative to merging-based approaches.

# B  ALGORITHM WORKFLOW

We formalize the complete workflow of our proposed framework in two distinct algorithms. **Algorithm 1** details the offline indexing phase, where `DocPruner` generates a compact set of document embeddings by adaptively pruning patches based on their attention-derived importance scores. Subsequently, **Algorithm 2** illustrates the online retrieval phase, where the final relevance score is efficiently computed via a `MaxSim` operation using this pruned set of embeddings.

---

**Algorithm 1:** The `DocPruner` Adaptive Pruning (Offline Indexing Phase)

---

**Input:** A document page $d$;
A VLM encoder $\Phi(\cdot)$ that outputs patch embeddings and attention weights;
A sensitivity controller hyperparameter $k$.
**Output:** A pruned set of patch embeddings $\mathbf{D}'_d$.

```
/* Step 0:  VLM Forward Pass                                              */
```
$\{\mathbf{D}_d, \mathbf{A}^{(L)}\} \leftarrow \Phi(d)$  `// Extract embeddings` $\mathbf{D}_d = \{\mathbf{d}_j\}_{j=1}^{L_d}$ `and final layer attention`
   $\mathbf{A}^{(L)}$

```
/* Step 1:  Quantifying Patch Importance                                  */
```
Let $g$ be the index of the global token (e.g., `[EOS]`)
Initialize an empty list of importance scores $\mathcal{I}_d$
**for** $j \leftarrow 1$ **to** $L_d$ **do**
  $\quad \bar{\mathbf{A}}_{g,j}^{(L)} \leftarrow \frac{1}{H} \sum_{h=1}^{H} \mathbf{A}_{h,g,j}^{(L)}$
  $\quad I(\mathbf{d}_j) \leftarrow \bar{\mathbf{A}}_{g,j}^{(L)}$     `// Importance is attention to patch` $j$ `(Eq. 2)`
  $\quad$ Append $I(\mathbf{d}_j)$ to $\mathcal{I}_d$
**end**

```
/* Step 2:  Adaptive Thresholding                                         */
```
$\mu_d \leftarrow \frac{1}{L_d} \sum_{j=1}^{L_d} I(\mathbf{d}_j)$     `// Calculate mean importance (Eq. 3)`
$\sigma_d \leftarrow \sqrt{\frac{1}{L_d} \sum_{j=1}^{L_d} (I(\mathbf{d}_j) - \mu_d)^2}$     `// Calculate std dev of importance (Eq. 4)`
$\tau_d \leftarrow \mu_d + k \cdot \sigma_d$     `// Define the document-specific threshold`

$\hat{\mathbf{D}}'_d \leftarrow \{\}$     `// Initialize preliminary pruned set`
**for** $j \leftarrow 1$ **to** $L_d$ **do**
  $\quad$ **if** $I(\mathbf{d}_j) > \tau_d$ **then**
  $\quad\quad \hat{\mathbf{D}}'_d \leftarrow \hat{\mathbf{D}}'_d \cup \{\mathbf{d}_j\}$     `// Keep patch if importance > threshold (Eq. 5)`
  $\quad$ **end**
**end**

```
/* Step 3:  Finalizing with Robustness Guarantee                          */
```
**if** $\hat{\mathbf{D}}'_d = \emptyset$ **then**
  $\quad j^* \leftarrow \underset{j \in \{1,\ldots,L_d\}}{\arg\max} \ I(\mathbf{d}_j)$
  $\quad \mathbf{D}'_d \leftarrow \{\mathbf{d}_{j^*}\}$     `// Keep the single most important patch (Eq. 6)`
**else**
  $\quad \mathbf{D}'_d \leftarrow \hat{\mathbf{D}}'_d$     `// Use the preliminary pruned set (Eq. 6)`
**end**

**return** $\mathbf{D}'_d$

---

---

**Algorithm 2:** Scoring with Pruned Embeddings (Online Retrieval Phase)

---

**Input:** A textual query $q$;
The pruned document embedding set $\mathbf{D}'_d$ (from Algorithm 1);
A VLM encoder $\Phi(\cdot)$ for query encoding.
**Output:** The relevance score $s'(q, d)$.

```
/* Step 1:  Encode Query                                          */
```
$\mathbf{Q} \leftarrow \Phi(q)$             `// Encode q into token embeddings` $\mathbf{Q} = \{\mathbf{q}_i\}_{i=1}^{L_q}$

```
/* Step 2:  Compute Score with Pruned Embeddings                  */
```
$s'(q, d) \leftarrow 0$
**for** $\mathbf{q}_i \in \mathbf{Q}$ **do**
    max_sim $\leftarrow -\infty$
    **for** $\mathbf{d}_j \in \mathbf{D}'_d$ **do**
        sim $\leftarrow \mathbf{q}_i^\top \mathbf{d}_j$
        **if** *sim > max_sim* **then**
            max_sim $\leftarrow$ sim
        **end**
    **end**
    $s'(q, d) \leftarrow s'(q, d) +$ max_sim     `// Aggregate max similarity per query token (Sec 3.2.3)`
**end**

**return** $s'(q, d)$

---

## C    DETAILS OF BENCHMARKS

This section provides detailed descriptions of the benchmarks used in our evaluation to validate the performance of `DocPruner`.

### C.1    VIDORE-V2 BENCHMARK

The ViDoRe-V2 benchmark (Macé et al., 2025) was designed to address the saturation of its predecessor, ViDoRe-V1 (Faysse et al., 2024), where top models were achieving near-perfect scores. It introduces more realistic and challenging retrieval scenarios by incorporating several key features: (1) **Blind Contextual Querying**, where query generation models have limited context, forcing them to create non-extractive questions that better mimic real user behavior; (2) **Long and Cross-Document Queries**, which require models to retrieve information from multiple pages or across different documents; and (3) a **Hybrid Generation Process**, combining synthetic query generation with extensive human-in-the-loop filtering to ensure high query quality. The benchmark comprises four diverse datasets: esg-reports-v2[2], biomedical-lectures-v2[3], economics-reports-v2[4], and esg-reports-human-labeled-v2[5], making it a robust testbed for model generalization.

Illustration of visual document examples from ViDoRe-V2 benchmark (Macé et al., 2025) can be seen in Figures 6, 7, and 8.

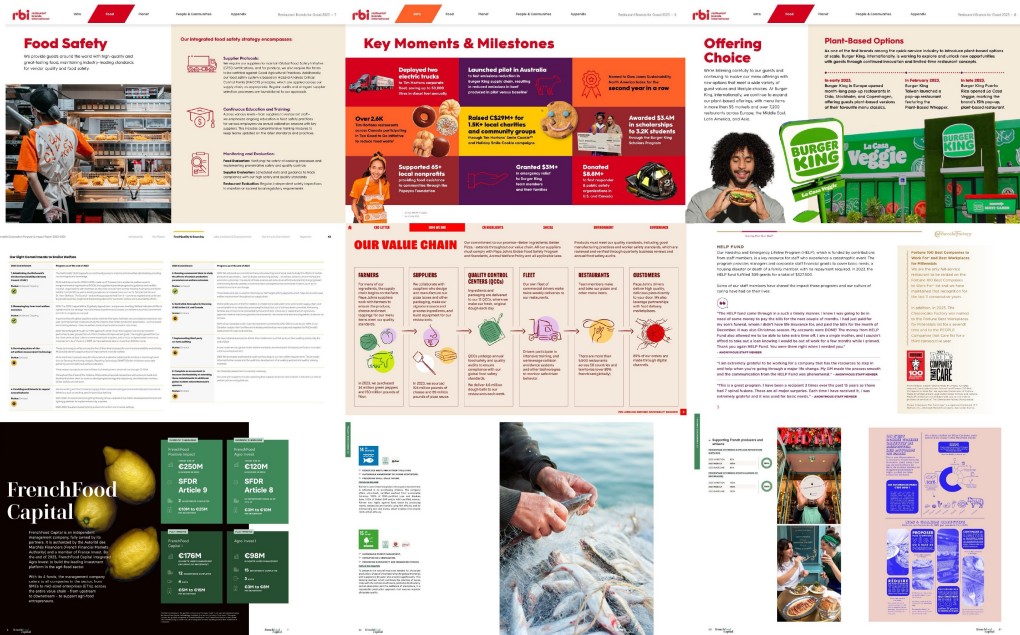

**Figure 6:** Illustration of visual document examples from *ESG* and *ESG-human* datasets (The latter is fully labelled by hand, and has no overlap of queries with its synthetic counterpart). They focus on the theme of **ESG reports from the fast food industry**.

---

[2]https://huggingface.co/datasets/vidore/esg_reports_v2
[3]https://huggingface.co/datasets/vidore/biomedical_lectures_v2
[4]https://huggingface.co/datasets/vidore/economics_reports_v2
[5]https://huggingface.co/datasets/vidore/esg_reports_human_labeled_v2

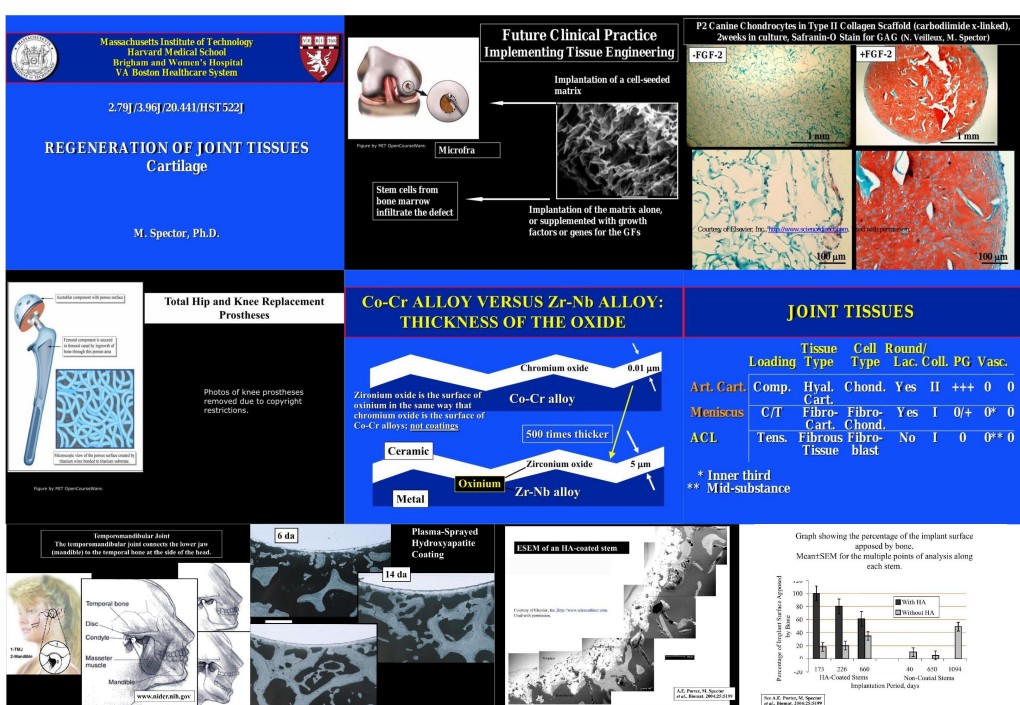

**Figure 7:** Illustration of visual document examples from *Biomedical Lectures* datasets. It focuses on the theme of **MIT courses in anatomy** (precisely tissue interactions).

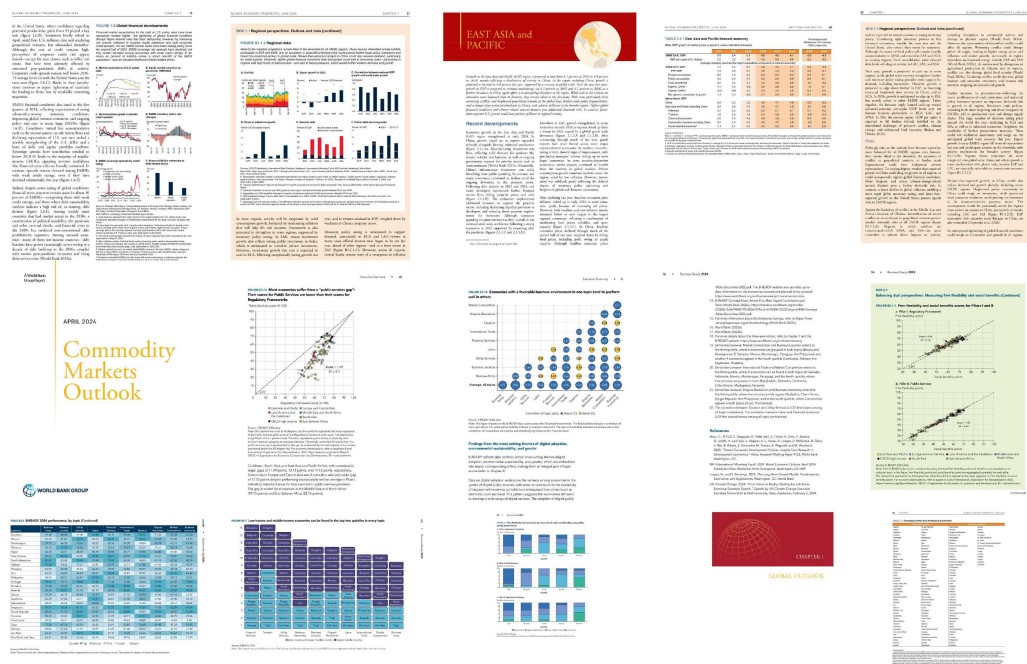

**Figure 8:** Illustration of visual document examples from *Economics Reports* datasets. It focuses on the theme of **World economic reports from 2024**.

## C.2 JINAVDR-BENCH

JinaVDR-Bench was introduced alongside Jina Embeddings v4 (Günther et al., 2025) to evaluate a new generation of unified embedding models capable of producing both single-vector (dense) and multi-vector representations from a single architecture. The benchmark is notable for its breadth, covering a wide array of document types and retrieval tasks. Its datasets include academic papers (Astro-ph), financial reports (DocILE, DeepForm), presentation slides (SlideVQA), technical manuals, and infographics (InfographicsVQA), among others. This diversity tests a model's ability to handle documents with varying layouts, languages (it includes multilingual splits), and content (*e.g.,* text-heavy, table-rich, or figure-dominant). By providing a standardized evaluation across these heterogeneous sources, JinaVDR-Bench serves as a comprehensive tool for assessing the versatility and robustness of VDR models. To evaluate the multilingual generalization of `DocPruner`, we choose europeana-de-news[6], beverages-catalogue-ru[7], shanghai-master-plan[8], and automobile-catalogue-jp[9] for German, Russian, Chinese, and Japanese visual documents, respectively.

Illustration of visual document examples from JinaVDR-Bench (Günther et al., 2025) can be seen in Figures 9, 10, 11, and 12.

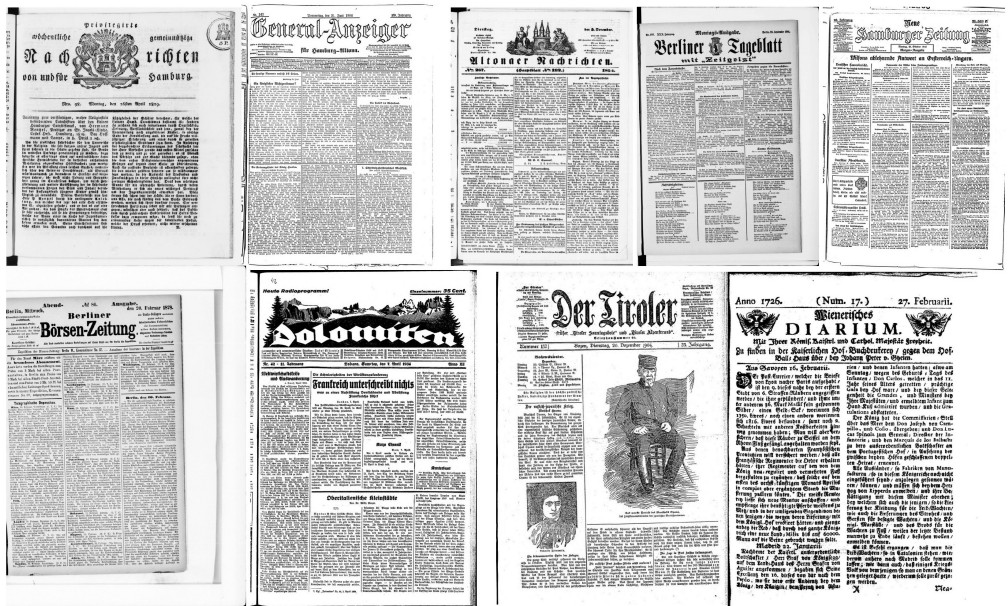

**Figure 9:** Illustration of visual document examples from *German* datasets. It focuses on the records of the European online collection by selecting scans of **German news articles**.

## C.3 VISUALIZATION OF PRUNED DOCUMENT CASES

This section illustrate the visual examples of pruned documents, as depicted by Figures 13 and 14.

---

[6] https://huggingface.co/datasets/jinaai/europeana-de-news
[7] https://huggingface.co/datasets/jinaai/beverages_catalogue_ru
[8] https://huggingface.co/datasets/jinaai/shanghai_master_plan
[9] https://huggingface.co/datasets/jinaai/automobile_catalogue_jp

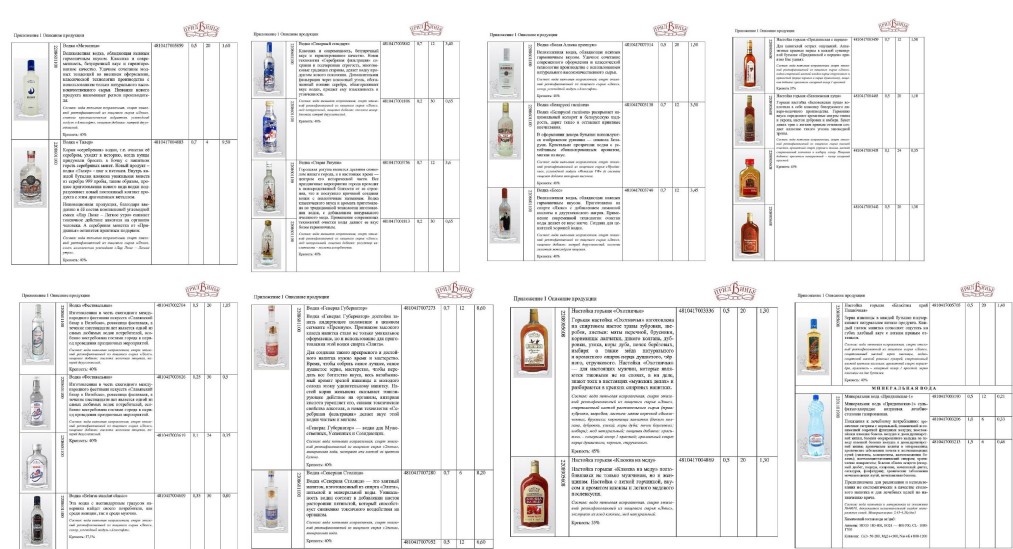

**Figure 10:** Illustration of visual document examples from *Russian* datasets. It focuses on the **beverage catalogs** on Google search and downloading PDFs.

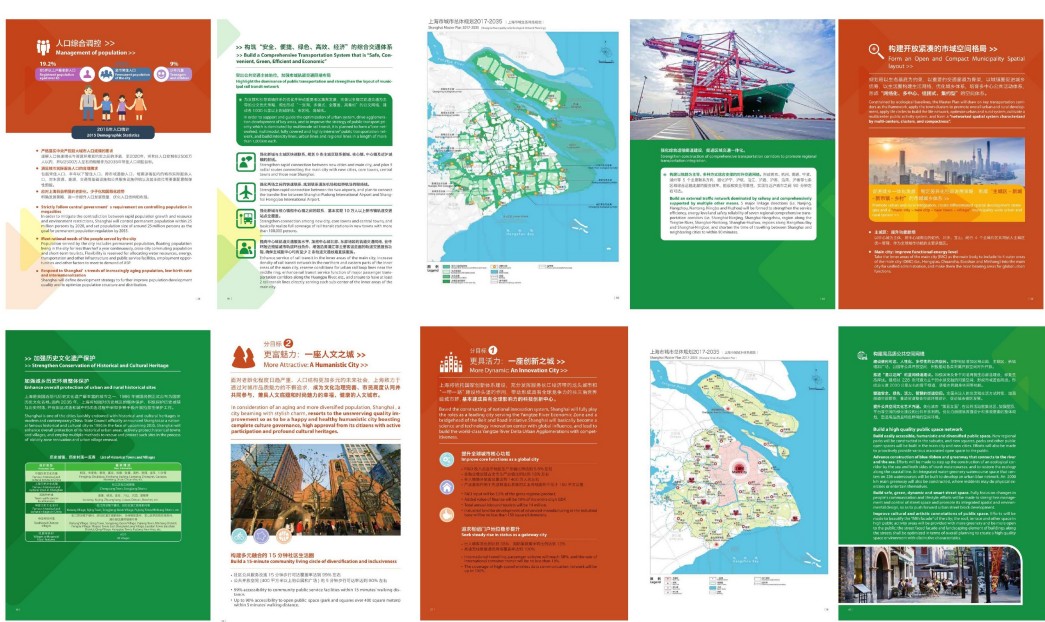

**Figure 11:** Illustration of visual document examples from *Chinese* datasets. It focuses on the theme of **Shanghai master plan document** taken from (Shanghai Municipal People's Government Urban Planning and Land Resource Administration Bureau, 2018).

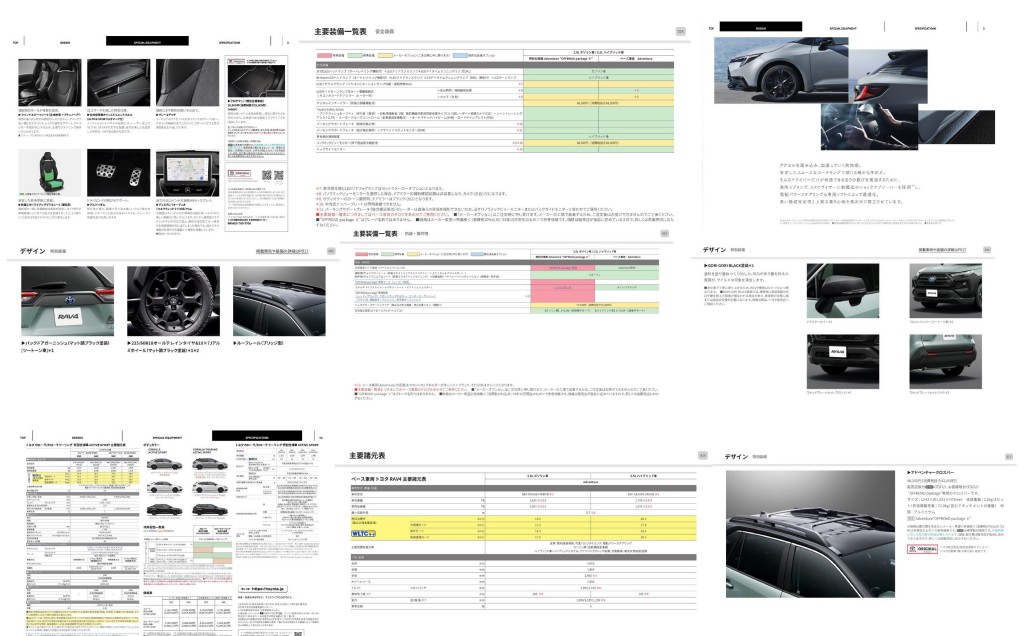

**Figure 12:** Illustration of visual document examples from *Japanese* datasets. It focuses on the theme of **marketing document** from Toyota Japanese website.

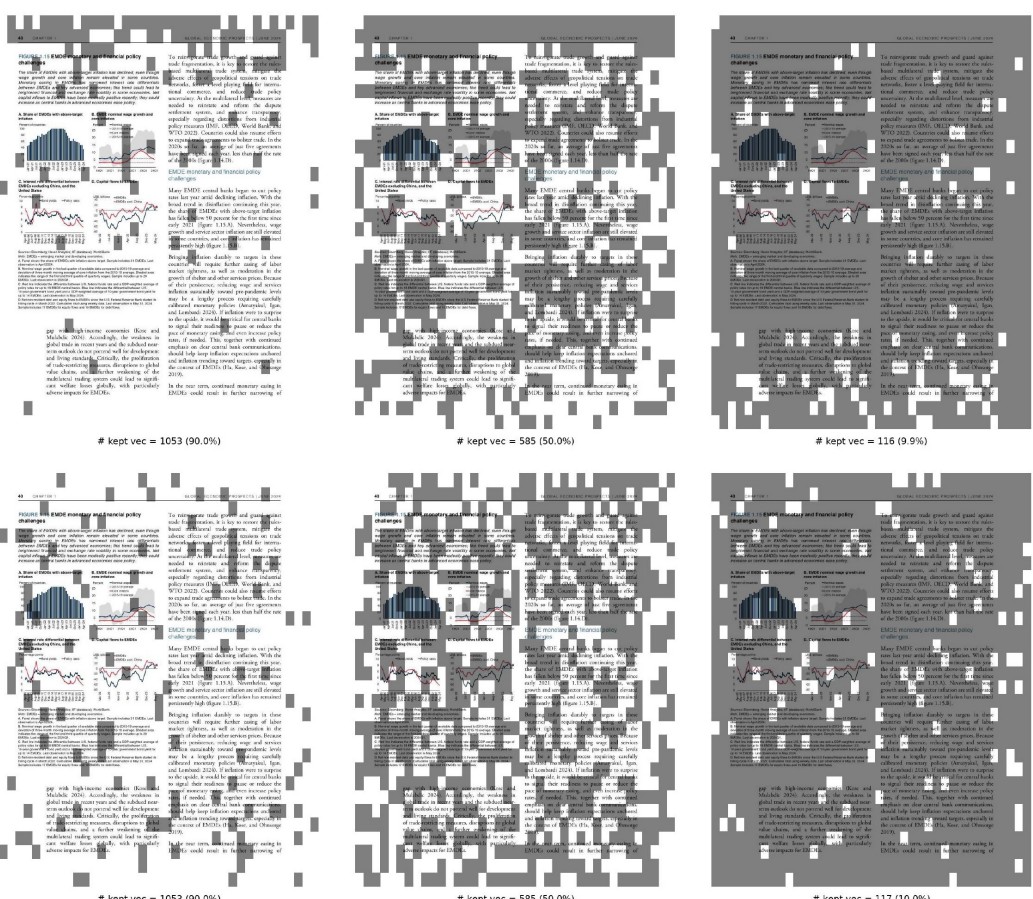

**Figure 13:** Illustration of visual document example one. Comparison of fixed-ratio pruning (*top row*) and random pruning (*bottom row*) at ratios of 10%, 50%, and 90% (*from left to right*). Gray patches are pruned and excluded from the multi-vector computation.

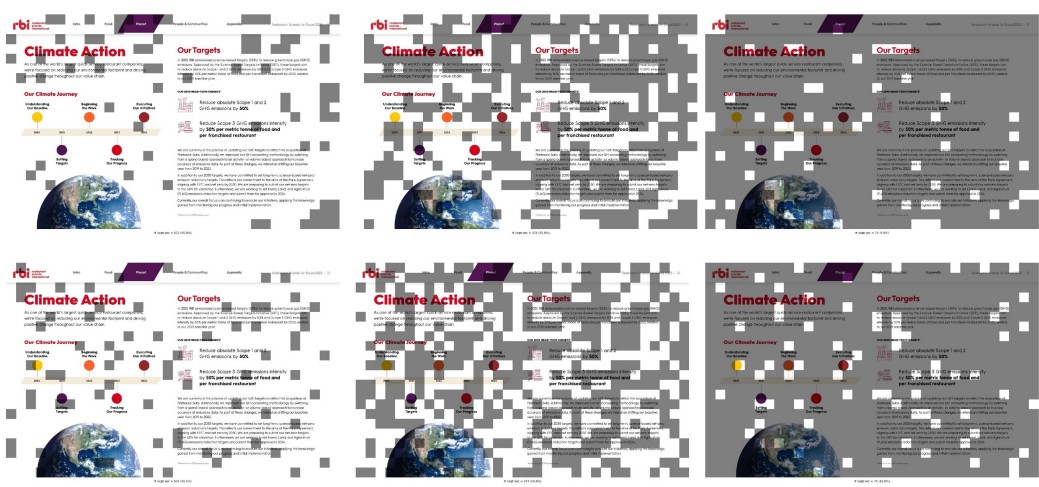

**Figure 14:** Illustration of visual document example two. Comparison of fixed-ratio pruning (*top row*) and random pruning (*bottom row*) at ratios of 10%, 50%, and 90% (*from left to right*). Gray patches are pruned and excluded from the multi-vector computation.

# D   DETAILS OF BASELINES

This section provides a detailed description of the implementation logic and hyperparameter settings for the baseline methods evaluated in Section 4.1. For each baseline, we empirically explored the specified hyperparameter space and selected the configurations that yielded the most representative performance trade-offs for presentation in our main results.

## D.1   MERGING-BASED METHODS

### D.1.1   SEM-CLUSTER

**Implementation Logic.**   This method performs semantic merging of patch embeddings. For each document, it first normalizes all patch embeddings. Then, it computes a pairwise distance matrix based on cosine similarity (`distance = 1 - cosine_similarity`). Using this matrix, it applies hierarchical agglomerative clustering with the 'ward' linkage method. The total number of patch embeddings is reduced by a *merging factor*, which determines the target number of clusters (*i.e.,* `num_clusters = num_patches / merging_factor`). Finally, the embeddings within each resulting cluster are averaged to produce a single centroid embedding, forming the new, smaller set of representations for the document.

**Hyperparameters.**

- **Merging Factor:** Defines the ratio by which the number of patch embeddings is reduced. A higher factor results in fewer clusters and thus more aggressive merging.
  - *Selection Range:* $\{2, 4, 9, 16, 25\}$.

### D.1.2   1D-POOLING

**Implementation Logic.**   This strategy treats the patch embeddings as a 1D sequence. It groups consecutive embeddings into non-overlapping windows of size equal to the *merging factor*. If the total number of patches is not divisible by the factor, the sequence is padded with zero vectors to ensure complete windows. The embeddings within each window are then averaged to create a single merged embedding. This effectively downsamples the sequence of patch embeddings.

**Hyperparameters.**

- **Merging Factor:** Specifies the size of the pooling window, *i.e.,* the number of sequential patch embeddings to be averaged into one.
  - *Selection Range:* $\{2, 4, 9, 16, 25\}$.

### D.1.3   2D-POOLING

**Implementation Logic.**   This method assumes a spatial arrangement of patches. The patch embeddings are first organized into a 2D grid that approximates their original spatial layout in the document image. This grid is padded with zero vectors to ensure its dimensions are divisible by the pooling kernel size. A 2D average pooling operation is then applied. The *merging factor*, which must be a perfect square, defines the area of the pooling window (*e.g.,* a factor of 4 corresponds to a 2x2 kernel). A mask is used during pooling to correctly normalize the averages, ensuring that padded areas do not contribute to the final merged embeddings.

**Hyperparameters.**

- **Merging Factor:** Defines the area of the 2D pooling window.
  - *Selection Range:* $\{4, 9, 16, 25\}$.

### D.2 Pruning-based Methods

#### D.2.1 Random

**Implementation Logic.** This naive baseline discards patch embeddings without considering their content. For each document, a specified *pruning ratio* of the total patch embeddings are selected uniformly at random and removed from the set. To ensure at least one patch remains, the implementation prevents pruning all patches even if the ratio is 1.0. This serves as a fundamental benchmark to gauge the performance loss from non-informed pruning.

**Hyperparameters.**

- **Pruning Ratio:** A float between 0.0 and 1.0 that specifies the fraction of patch embeddings to be randomly discarded.
  - *Selection Range:* $\{0.1, 0.3, 0.5, 0.7, 0.9\}$.

#### D.2.2 Attention-plus-Similarity

**Implementation Logic.** This adaptive method computes a composite score for each patch to decide whether to prune it. The score is a weighted sum of two components: (1) an **importance score**, derived from the attention weight the global `[EOS]` token pays to the patch, and (2) a **representativeness score**, calculated as the cosine similarity between the patch embedding and the `[EOS]` embedding. The final score is pruned using an adaptive threshold calculated as $\mu + k \cdot \sigma$, where $\mu$ and $\sigma$ are the statistics of the composite scores for that document. The results presented in the paper were based on an empirical grid search over all hyperparameter combinations, selecting the optimal $\alpha$ for $k = 0$ and $k = 1$ respectively to show representative results.

**Hyperparameters.**

- **Adaptation Factor** ($k$)**:** A coefficient that controls the strictness of the dynamic pruning threshold. A higher value leads to a more aggressive pruning.
  - *Selection Range:* $\{-0.5, -0.25, 0, 0.25, 0.5, 1\}$.
- **Weighting Factor** ($\alpha$)**:** A float between 0.0 and 1.0 that balances the contribution of the importance score (attention) and the representativeness score (similarity).
  - *Selection Range:* $\{0.1, 0.3, 0.5, 0.7, 0.9\}$.

#### D.2.3 Pivot-Threshold

**Implementation Logic.** This advanced adaptive baseline employs a two-stage pruning process. It first identifies an "important set" of patches by applying an adaptive attention-based threshold ($\mu + k \cdot \sigma$ of `[EOS]`-to-patch attention scores), similar to the core mechanism of `DocPruner`. Within this important set, it selects a fixed *pivot num* of patches as "pivots". For the remaining non-pivot patches in the important set, it calculates a duplication score, defined as the maximum cosine similarity to any of the pivots. A second adaptive threshold ($\mu_{\text{dup}} + k_{\text{dup}} \cdot \sigma_{\text{dup}}$ of these duplication scores) is then used to prune non-pivot patches that are deemed too similar to the pivots. We found $k_{\text{dup}} = 1$ and *pivot_num* $= 10$ were consistently optimal via empirical search. Therefore, the results presented fix these two hyperparameters and show the performance trade-off by varying the adaptation factor $k$.

**Hyperparameters.**

- **Adaptation Factor** ($k$)**:** Controls the threshold for initial importance-based filtering stage.
  - *Selection Range:* $\{-0.5, -0.25, 0, 0.25, 0.5, 1\}$.
- **De-duplication Factor** ($k_{\text{dup}}$)**:** Controls the similarity threshold for the second stage.
  - *Selection Range:* $\{-0.5, -0.25, 0, 0.25, 0.5, 1\}$.
- **Pivot Num:** The number of pivot tokens to select from the important set for the de-duplication stage.

– *Selection Range:* $\{5, 10, 15, 20\}$.

# E MORE EXPERIMENTAL ANALYSIS

## E.1 MORE EXPERIMENT ON VIDORE-V2

Performance comparison (nDCG@5) between `DocPruner` and baselines on ViDoRe-V2 benchmark across four datasets on **ColQwen2.5**, **ColNomic**, and **Jina Embedding V4** can be seen in Figures 15, 16, and 17, respectively. Pruning ratio distribution of `DocPruner` on **ColQwen2.5**, **ColNomic**, and **Jina Embedding V4** can be seen in Figures 18, 19, and 20, respectively.

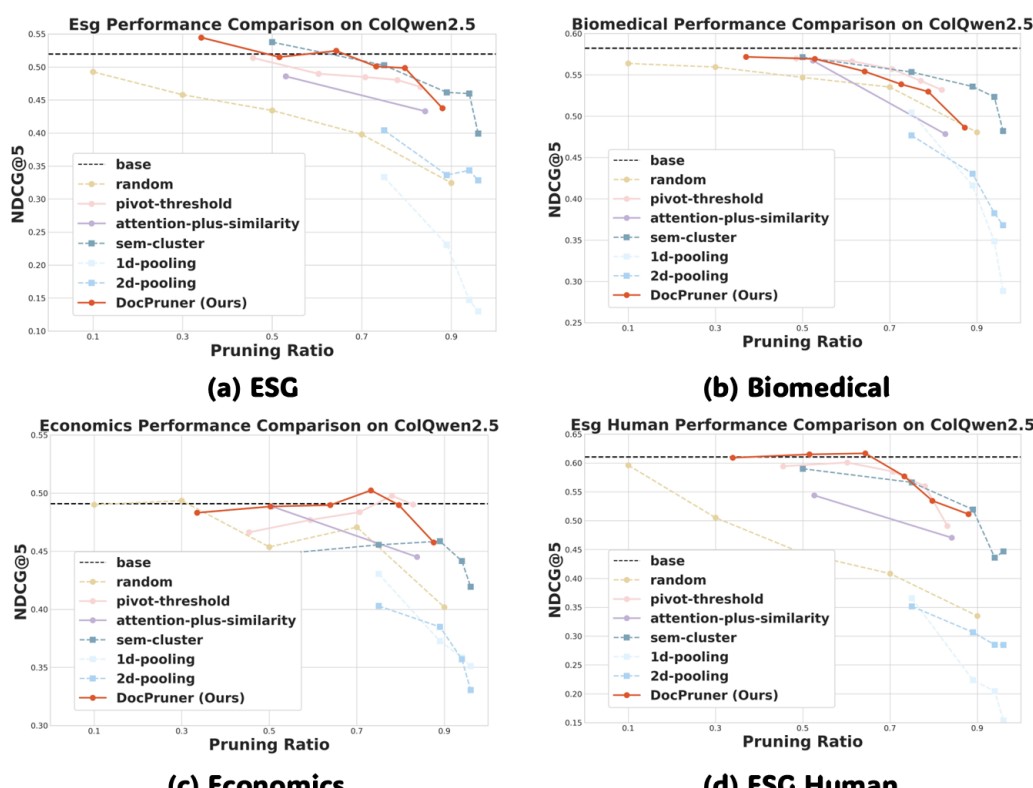

**(a) ESG**

**(b) Biomedical**

**(c) Economics**

**(d) ESG Human**

**Figure 15:** Performance comparison (nDCG@5) of **ColQwen2.5** between `DocPruner` and baselines on ViDoRe-V2 benchmark across four datasets.

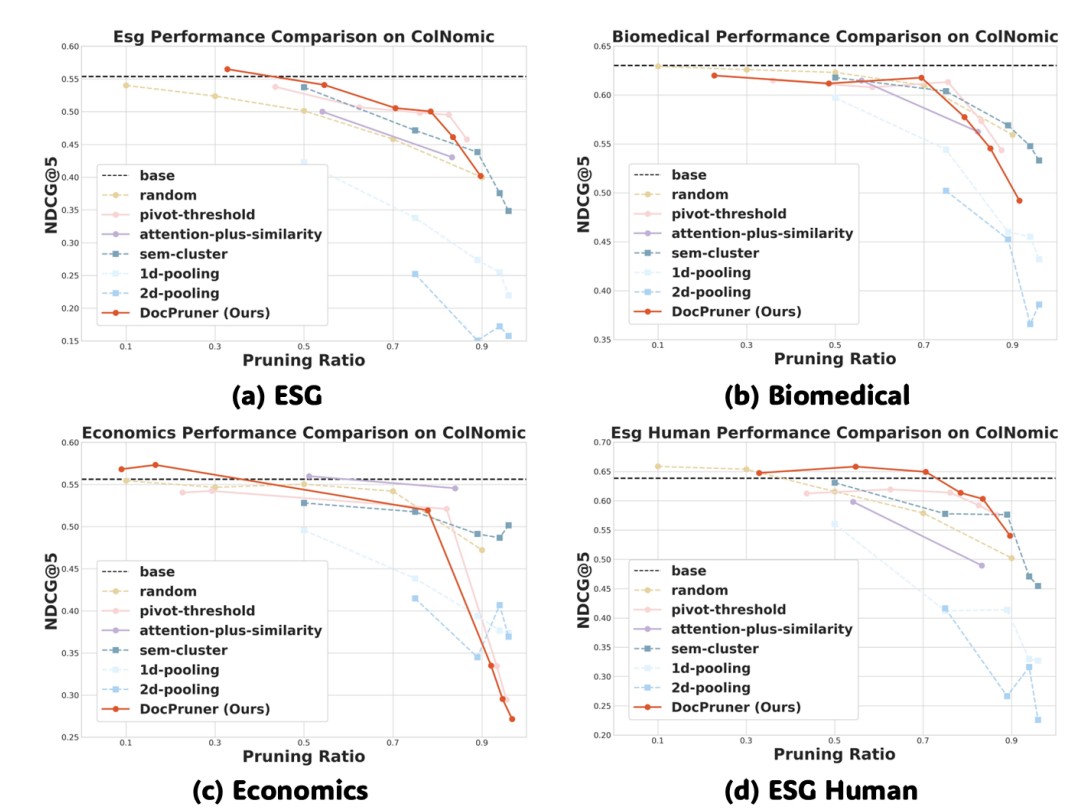

**Figure 16:** Performance comparison (nDCG@5) of **ColNomic** between `DocPruner` and baselines on ViDoRe-V2 benchmark across four datasets.

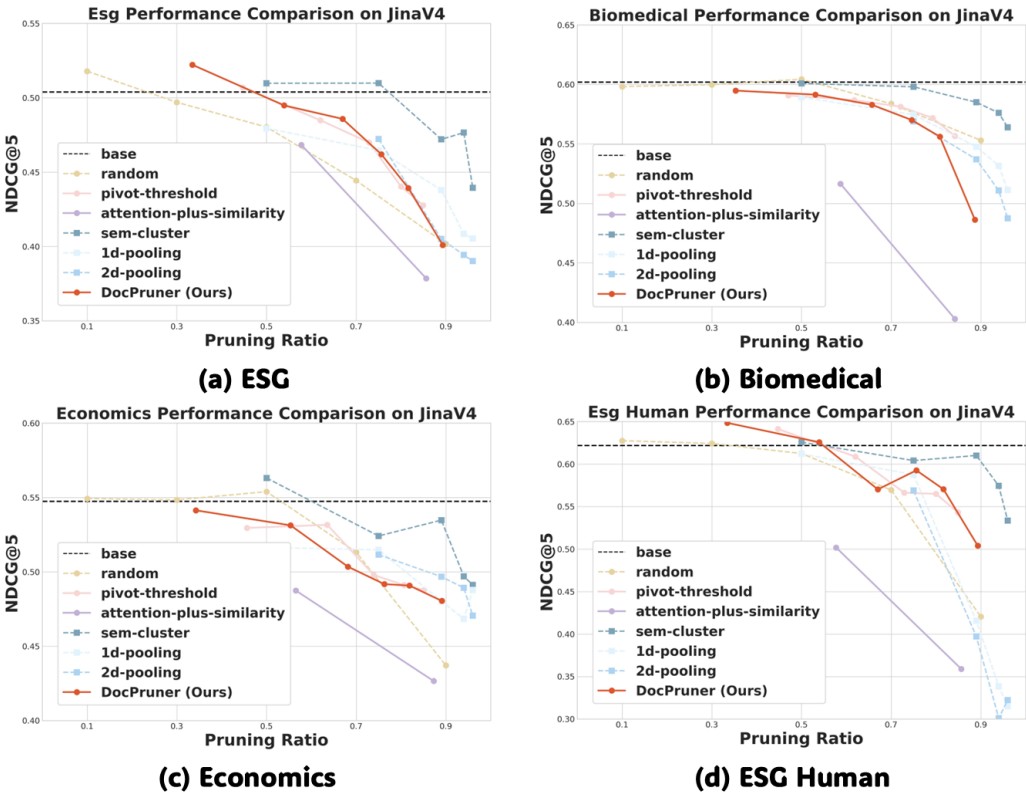

**Figure 17:** Performance comparison (nDCG@5) of **Jina Embedding V4** between `DocPruner` and baselines on ViDoRe-V2 benchmark across four datasets.

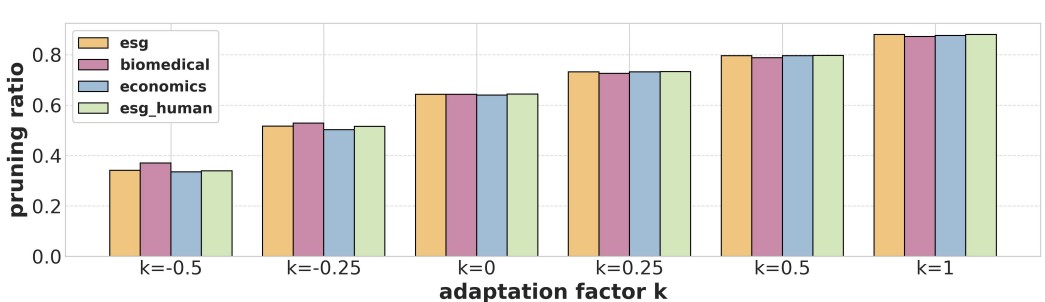

**Figure 18:** Pruning ratio distribution of **ColQwen2.5** using `DocPruner` across four datasets of ViDiRe-V2 over a *adaptation factor k* range of {-0.5, -0.25, 0, 0.25, 0.5, 1}.

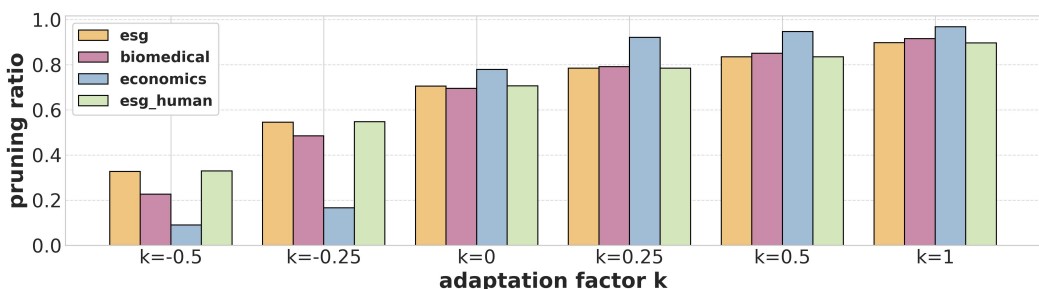

**Figure 19:** Pruning ratio distribution of **ColNomic** using `DocPruner` across four datasets of ViDiRe-V2 over a *adaptation factor k* range of {-0.5, -0.25, 0, 0.25, 0.5, 1}.

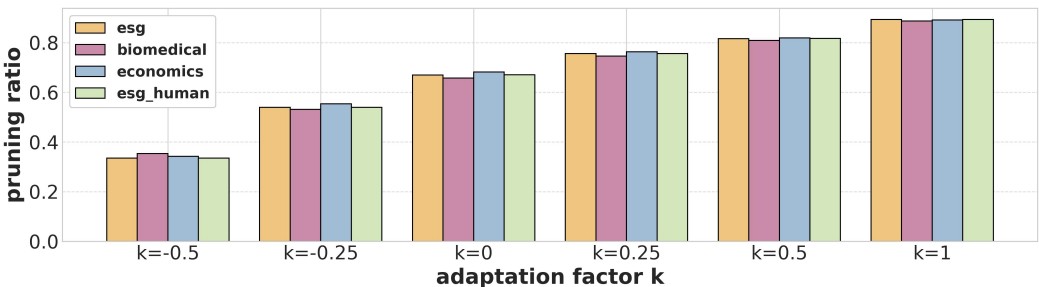

**Figure 20:** Pruning ratio distribution of **Jina Embedding V4** using `DocPruner` across four datasets of ViDiRe-V2 over a *adaptation factor k* range of {-0.5, -0.25, 0, 0.25, 0.5, 1}.

## E.2 MORE EXPERIMENT ON JINAVDR

Performance comparison (nDCG@5) between `DocPruner` and baselines on JinaVDR benchmark across four multilingual datasets on **ColQwen2.5**, **ColNomic**, and **Jina Embedding V4** can be seen in Figures 21, 22, and 23, respectively.

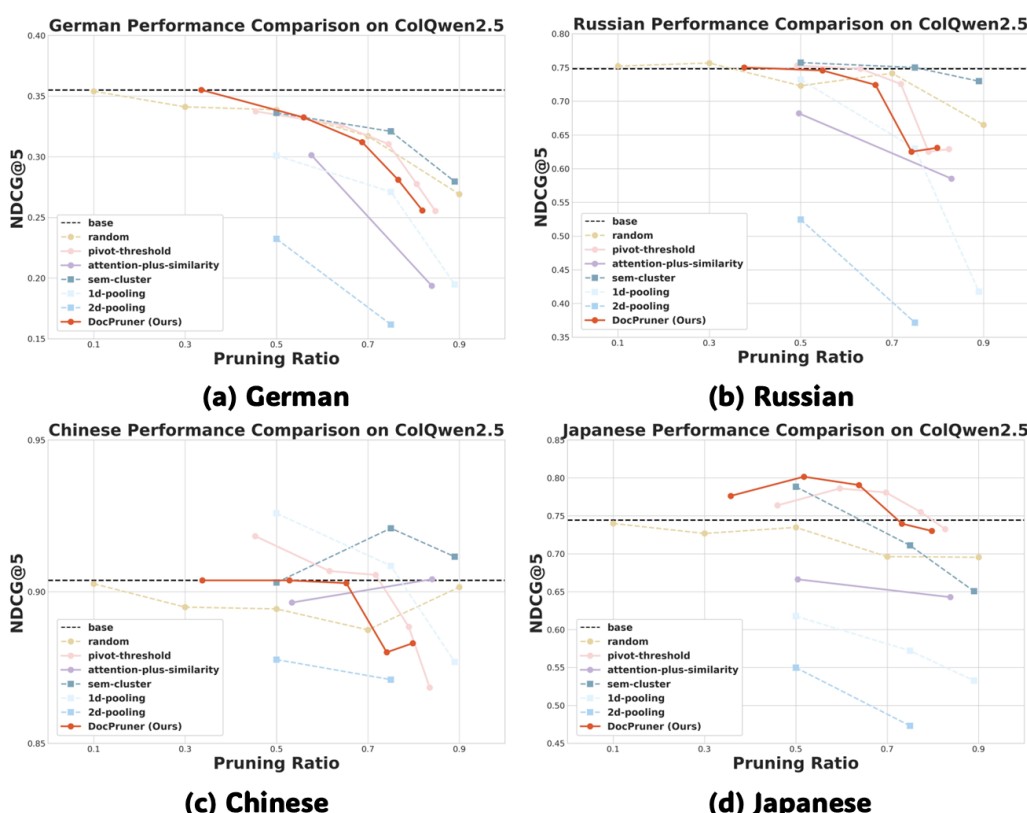

**Figure 21:** Performance comparison (nDCG@5) of **ColQwen2.5** between `DocPruner` and baselines on Ji-naVDR benchmark across four datasets.

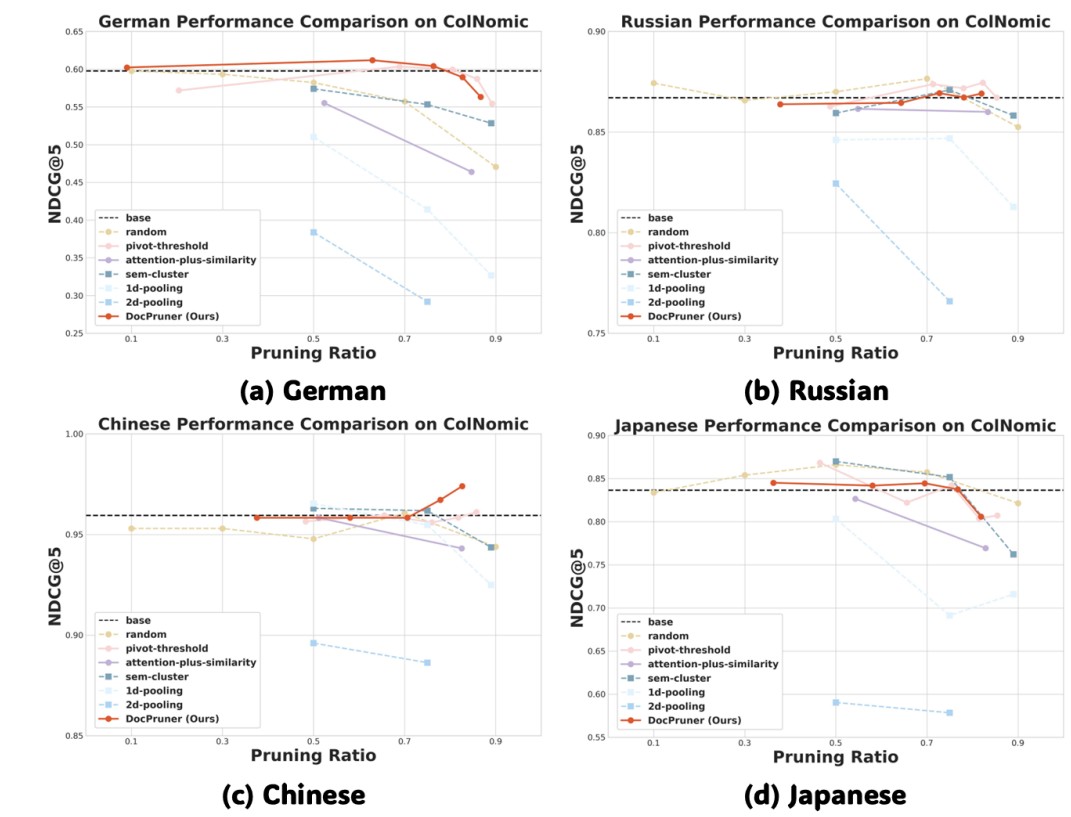

**Figure 22:** Performance comparison (nDCG@5) of **ColNomic** between `DocPruner` and baselines on JinaVDR benchmark across four datasets.

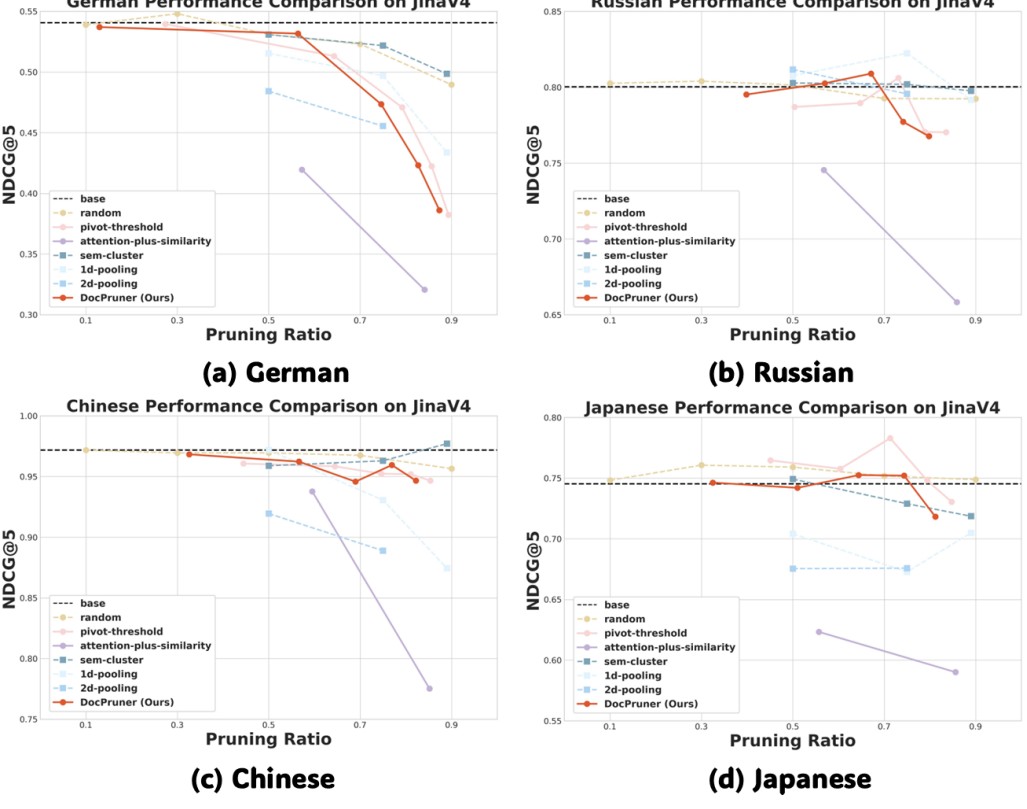

**Figure 23:** Performance comparison (nDCG@5) of **Jina Embedding V4** between `DocPruner` and baselines on JinaVDR benchmark across four datasets.

## E.3 MORE VARIANT STUDY

Performance comparison (nDCG@5) between `DocPruner` and other variants on ViDoRe-V2 benchmark across four datasets on **ColQwen2.5**, **ColNomic**, and **Jina Embedding V4** can be seen in Figure 24. The prompt used for evaluating `attention-threshold-nfp` is shown below.

> **Prompt Template for `attention-threshold-nfp`**
>
> ```
> Analyze this document page.  Assign high importance to
> regions containing text, tables, charts, and meaningful
> figures.  Assign low importance to decorative graphics,
> logos, empty space, and repeating headers or footers.
> ```

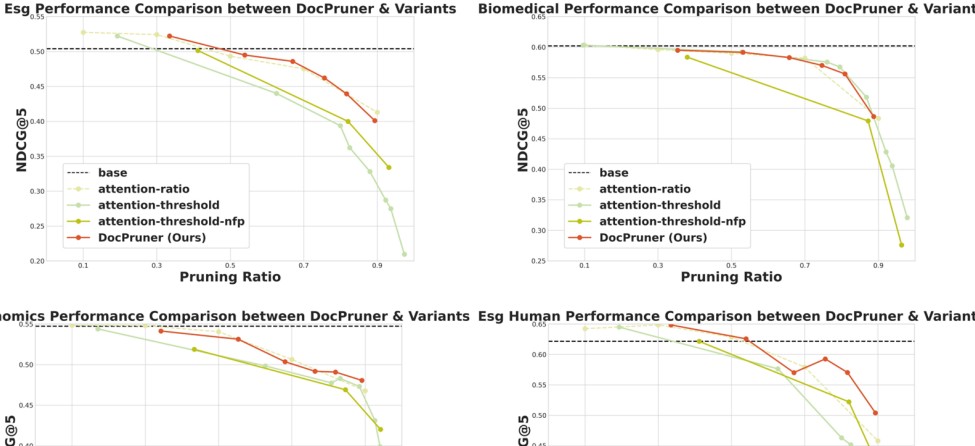

**Figure 24:** Performance comparison between `DocPruner` and other variants on ViDoRe-V2 benchmark across four datasets.

### E.4 MORE ANALYSIS OF ADAPTATION FACTOR

The adaptation factor, $k$, is the central hyperparameter in the `DocPruner` framework, governing the delicate trade-off between storage compression and retrieval performance. Its significance lies not in setting a fixed threshold, but in modulating a dynamic one, $\tau_d = \mu_d + k \cdot \sigma_d$, which is tailored to the unique statistical properties of each document's attention score distribution. As illustrated in Figures 25-28, the interaction between $k$ and the distribution's shape (characterized by its mean $\mu_d$ and standard deviation $\sigma_d$) is the key to `DocPruner`'s adaptive behavior.

Furthermore, we observe a consistent principle across different VDR models (ColQwen2.5, Jina Embedding V4, and ColNomic). While the absolute attention scores and the specific shapes of the distributions may differ, each model fundamentally learns to assign higher attention to semantically meaningful regions. `DocPruner`'s statistical approach is robust to these model-specific variations, as it operates on the relative distribution of scores within each document. This intrinsic, document-relative nature is why a single $k$ value (*e.g.,* k=-0.25) can yield a desirable balance of $\sim$50% compression and near-lossless performance across diverse models and datasets.

In summary, the adaptation factor $k$ is not a simple cutoff value but a sensitivity parameter. It leverages the statistical fingerprint of each document's attention distribution to determine the appropriate pruning intensity. This enables `DocPruner` to automatically apply a "one-size-fits-one" strategy: conservative pruning for information-rich pages and aggressive pruning for sparse ones, achieving a robust and efficient compression solution for real-world visual document retrieval.

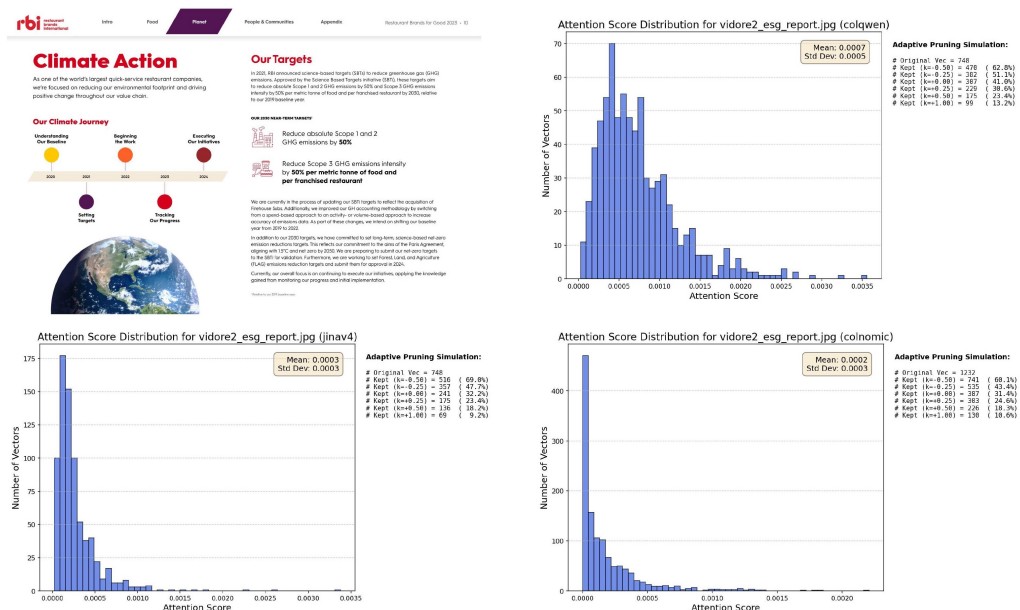

**Figure 25:** The attention score distribution and number of kept vectors (wrt. different adaption factor setting) of the representative document from *ESG* subset of ViDoRe-V2. The **raw document** is shown in top-left corner, and the rest represent histograms corresponding to **ColQwen2.5**, **Jina Embedding V4**, and **ColNomic**.

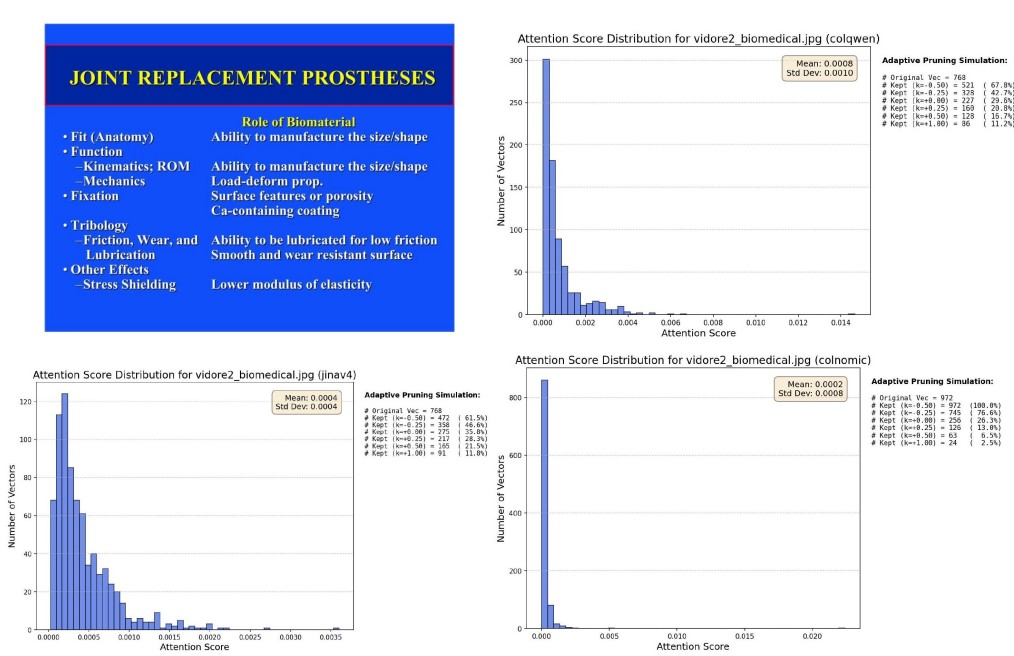

**Figure 26:** The attention score distribution and number of kept vectors (wrt. different adaption factor setting) of the representative document from *Biomedical* subset of ViDoRe-V2. The **raw document** is shown in top-left corner, and the rest represent histograms corresponding to **ColQwen2.5**, **Jina Embedding V4**, and **ColNomic**.

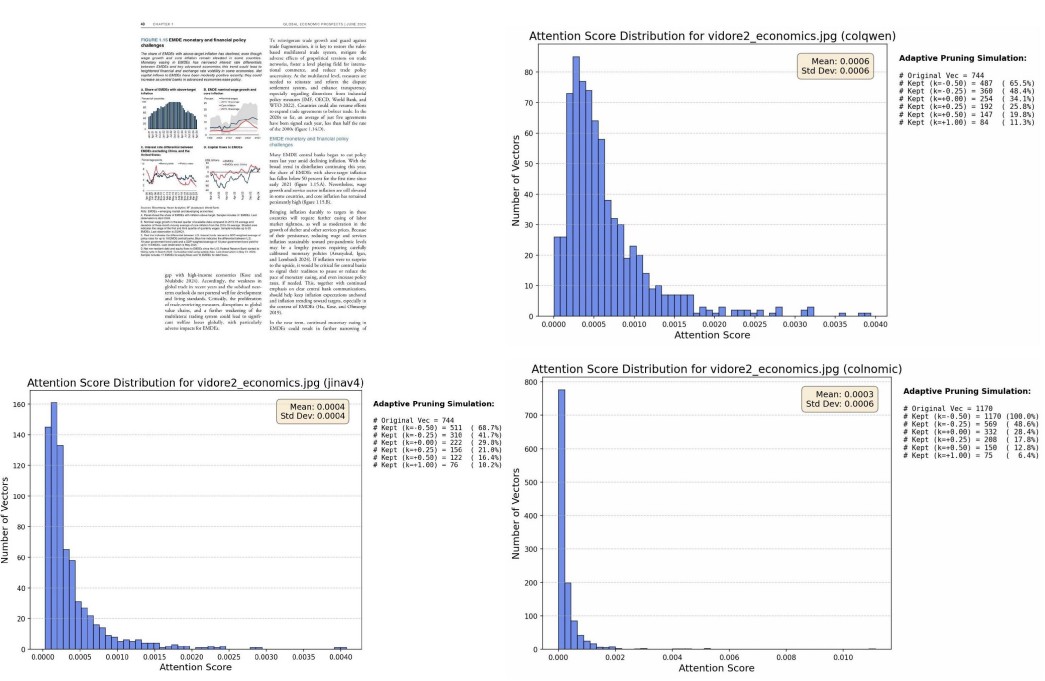

**Figure 27:** The attention score distribution and number of kept vectors (wrt. different adaption factor setting) of the representative document from *Economics* subset of ViDoRe-V2. The **raw document** is shown in top-left corner, and the rest represent histograms corresponding to **ColQwen2.5**, **Jina Embedding V4**, and **ColNomic**.

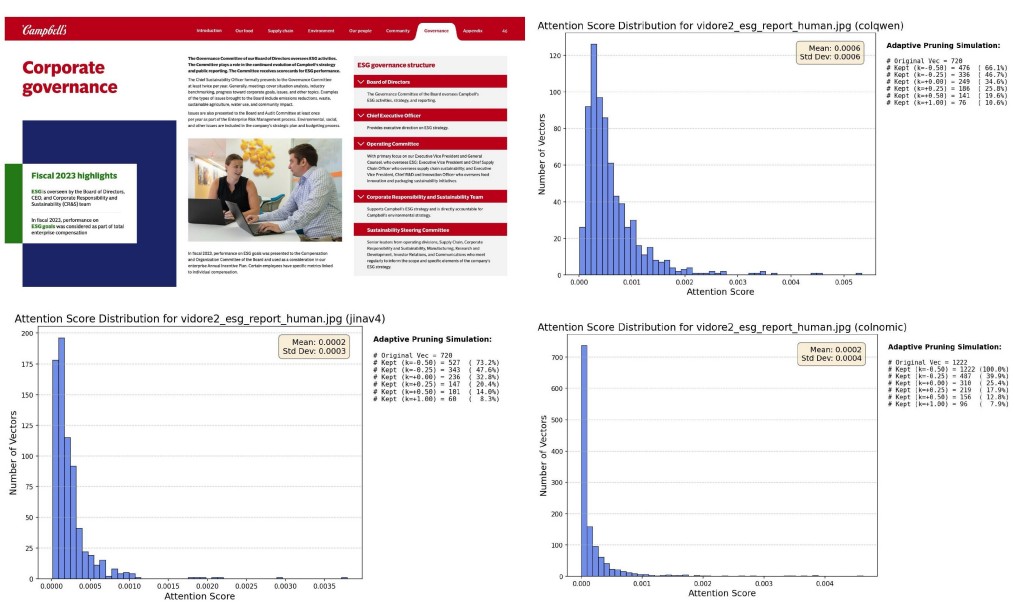

**Figure 28:** The attention score distribution and number of kept vectors (wrt. different adaption factor setting) of the representative document from *ESG Human* subset of ViDoRe-V2. The **raw document** is shown in top-left corner, and the rest represent histograms corresponding to **ColQwen2.5**, **Jina Embedding V4**, and **ColNomic**.

## F  BROADER IMPACT

The development of `DocPruner` carries significant positive impacts that extend from the research community to industrial applications and ultimately to society at large. `DocPruner` addresses a critical, practical bottleneck in state-of-the-art VDR, and its implications can be understood on three distinct levels.

First, within the **academic and research community**, `DocPruner` encourages a paradigm shift. While much of the recent focus has been on scaling up models to achieve marginal gains in accuracy, our work highlights the paramount importance of computational and storage efficiency. By providing a simple yet effective framework for making powerful multi-vector models practical, we hope to inspire more research into resource-aware AI. This can enable researchers, particularly those in resource-constrained environments, to conduct larger-scale experiments and explore more complex VDR tasks that were previously computationally prohibitive. Our work serves as a proof-of-concept that "smarter" resource management can be as impactful as "bigger" models.

Second, for **industry and commercial applications**, `DocPruner` offers a direct and substantial economic benefit. The prohibitive storage costs associated with multi-vector embeddings are a major barrier to the widespread adoption of advanced VDR systems in enterprise settings. By reducing storage requirements by 50-60% with negligible performance loss, `DocPruner` makes it economically feasible for businesses in sectors like legal, finance, healthcare, and e-commerce to deploy high-fidelity document search and analysis tools. This can unlock new efficiencies in knowledge management, accelerate workflows that rely on searching vast archives of visually-rich documents (*e.g.,* contracts, financial reports, patent filings), and ultimately democratize access to state-of-the-art retrieval technology for a wider range of organizations.

Finally, on a broader **societal level**, the principles behind `DocPruner` contribute to making information more accessible and discoverable. Public institutions such as libraries, museums, and government archives are custodians of immense collections of digitized historical and cultural documents. The ability to affordably index and search these visual archives at a fine-grained level can empower educators, historians, and the general public, fostering new avenues for research and learning. By lowering the technological and financial barriers to building powerful search systems, our work can help preserve and unlock the value latent within our collective cultural and scientific heritage, contributing to a more informed and connected society.

## G  THE USE OF LARGE LANGUAGE MODELS (LLMS)

In the preparation of this manuscript, we utilized LLMs primarily for the purpose of language refinement and improving the clarity and readability of our writing. The LLM was employed as an editing assistant to help polish sentence structures, correct grammatical errors, and ensure consistency in terminology. The core scientific ideas, the design and implementation of the `DocPruner` framework, the execution of experiments, and the analysis of the results were all conceived and conducted exclusively by the authors. The LLM's role was strictly limited to enhancing the quality of the English prose and did not contribute to the intellectual content of the research.

