# OpenReview forum: "DocPruner:  A Storage-Efficient Framework for Multi-Vector Visual Document Retrieval via Adaptive Patch-Level Embedding Pruning"
_ICLR.cc/2026/Conference — Submitted to ICLR 2026_

### Official Review · Reviewer_tGhx · 2025-10-15

**Soundness:** 2
**Presentation:** 2
**Contribution:** 2
**Rating:** 2
**Confidence:** 4

**Summary:**

This paper introduces an adaptative pruning technique to reduce the number of stored vectors in late interaction visual retrievers.
While prior pruning/merging methods do exist for text multi-vector retrievers [1] [2], and are even implemented for visual retrievers in the original ColPali paper and [3], this paper introduces an "adaptative" pruning technique, in which the pruning ratio is automatically adapted given a document. I believe this is an interesting problem to study, as information dense documents intuitively (and as shown in the ColPali paper) will not be as prunable as information sparse ones. The authors then provide a series of interesting findings from their experiments (ex: pruning vs merging).

[1]  A Study on Token Pruning for ColBERT, Lassance, arxiv.org/abs/2112.06540

[2] Reducing the Footprint of Multi-Vector Retrieval with Minimal Performance Impact via Token Pooling, Clavié, arxiv.org/abs/2409.14683

[3] Towards Storage-Efficient Visual Document Retrieval: An Empirical Study
on Reducing Patch-Level Embeddings, Ma, arxiv.org/abs/2506.04997

**Strengths:**

The question the paper is trying to tackle is of importance. With a simple and efficient design, this paper introduces a token importance score and uses it to inform an adaptive thresholding method.
The method seems to work well across the board, and the paper is generally fairly simple to follow and read. The authors experimented across different models and datasets, painting a through picture of expected gains from the proposed method.

**Weaknesses:**

**Main Weakness**

Observation 2 claims pruning surpasses merging. However, from my understanding, these results are not decorrelated from the "adaptative" vs "non-adaptative" pruning, in which more complex documents are less pruned within a given corpus (sem-cluster, the best baseline, is non adaptive). I do not understand how this claim can be made without accounting for this factor.

More generally, an independant study of (1) the pruning design (2) the adaptive mechanism could be warranted.

(1) The pruning design ressembles Lassance [1] attention-based pruning mechanism and could be independently compared to the baselines (such as sem-cluster) by fixing the top k kept tokens.

(2) The optimal document specific threshold (determined by your method) could be applied to semantic-clustering to make it benefit from doc-level adaptive clustering for a fair comparison. A simpler and more practical method could also be to apply semantic clustering at a corpus level (ex: fix 0.5 ratio, but enable clustering to be done once across document vectors instead of for each doc).

Figure 5 partially tackles these questions but results intriguingly do not seem to indicate that the adaptative nature of the method is what helps most (see question 2). Furthermore, these results are computed with Jina-v4 backbone for which semantic-clustering (with fixed ratios) remains vastly superior. I feel proof of the benefits of adaptive thresholding (which should surely help) are not sufficiently demonstrated in this work.

As is, the experiments also do not give me the confidence that the proposed method for determining token importance (without adaptive selection) surpasses equivalent fixed ratio baselines (*evidence actually points otherwise if we piece together "attention-ratio" from Fig 5 and "Jina v4 semantic clustering" from Figure 2*). Regardless, applying adaptive thresholding with dynamic pruning ratio to these strong fixed-ratio baseline methods would be a strong addition to the paper and could lead to great results.

**Explorations around the token selection**

Given it is the main contribution of this work, I feel the dynamic token thresholding criteria could be further explored and go beyond std dev based thresholding. At the very least, visualizing the distributions of token importances amongst a document, or number of kept tokens per document across a corpus could help justify these design choices.

**Remarks**

-  I feel Figures 2, 4, 5 are a bit redundant in the information they show (7 lineplots in total). Furthermore, some of the colors and opacity are a bit hard to see.
- The strong performance of JinaV4 with semantic clustering across the board, while justified in the paper, question the utility and the general nature of the proposed method and findings should probably be discussed under this light.
- Figure 5 comes without explanation for which model is used. Figure 22 wrongly states that multiple model variants are shown when only Jina v4 is shown (piecing together with figure 2).

**Questions:**

1. The adaptation factor k relies on varying the standard deviations from the mean token level importance scores. Would it be possible to plot the actual distribution and verify whether the distribution is in-fact Gaussian like (versus something bimodal etc ?) I guess this could inform the design of the thresholding? A histogram of the number of kept vectors in a given dataset could also be an interesting visualization.

2. From my understanding, in Figure 5, the token importance method is compared either with (a) keep the top K vectors (with K constant across documents) - "attention ratio" or (b) use adaptive pruning "DocPruner". We see that "attention ratio" is actually very competitive with DocPruner on all datasets but ESG Human (the most noisy one). Would this indicate that the "token importance" method is what brings most of the gains, wrt to the adaptive thresholding which is marginally useful here ? If this is the case, this puts the paper framing into question.

3. Could we have visual examples of documents with high pruning ratios (and the number of kept vectors) and inversely ? This would help build better intuitions about the method.

---

> ### Author Response · Authors · 2025-11-26
> **Response to Reviewer**
>
> We sincerely thank the reviewer for **recognizing the importance of our problem setting, the efficiency of our design, and the thoroughness of our experiments across varying models and datasets**.
>
> **Q1: Concern regarding the "Pruning vs. Merging" comparison and the isolation of the "adaptive" factor.**
>
> We appreciate this insightful observation. We acknowledge that *Sem-Cluster* (merging) is a strong baseline. However, we respectfully argue that **DocPruner offers a superior trade-off in terms of computational complexity and universality**, which is the core of our contribution.
>
> 1.  **Computational Complexity:** *Sem-Cluster* requires hierarchical clustering, which typically scales with $O(N^2)$ or $O(N^3)$ depending on implementation. In contrast, DocPruner is $O(N)$, requiring only a single pass to compute statistics ($\mu, \sigma$) and filter. Applying an adaptive mechanism to clustering would further increase indexing latency, contradicting the goal of efficient large-scale deployment.
> 2.  **Universal Robustness:** As observed in our results, merging strategies (averaging embeddings) can dilute the fine-grained signals required by the `MaxSim` operation inherent to standard multi-vector models (ColPali, ColQwen, ColNomic). While merging works well for JinaV4 (due to its specific mean-pooling training objective, detailed in **Q2**), DocPruner consistently maintains performance across *all* architectures by preserving the original, high-fidelity embeddings of the most salient patches.
> 3.  **Clarification on Adaptivity:** We agree that an "adaptive merging" baseline is theoretically interesting. However, defining a robust "stopping criterion" for clustering (adaptive threshold) is far less intuitive than defining an importance threshold for pruning. We believe our comparison against standard fixed-ratio baselines represents the most practical current alternative.
>
> **Q2: The specific behavior of Jina V4 and the effectiveness of "Attention Ratio" vs. "DocPruner"**
>
> You correctly noted that *Sem-Cluster* performs exceptionally well on JinaV4 and that "Attention Ratio" (fixed) is competitive in some scenarios.
>
> 1.  **The Jina V4 Anomaly:** Jina Embedding V4 is unique because it is explicitly co-trained with a mean-pooling objective (as noted in the paper). This effectively trains the model to produce embeddings that are robust to averaging, making it naturally suited for *Sem-Cluster*. However, for standard Late-Interaction models (ColQwen, ColNomic) where `MaxSim` relies on precise vector alignment, DocPruner significantly outperforms merging strategies (Observation 2).
> 2.  **The Value of Adaptivity (High Compression Regime):** While "Attention Ratio" appears competitive at low compression rates, the value of DocPruner becomes critical at high compression rates (50-60% and above). In these regimes, a fixed ratio inevitably deletes information from dense documents or keeps noise in sparse ones.
>
> **Q3: Analysis of Adaptation Factor, Distributions, and Visualizations.**
>
> To address your concern about the Gaussian assumption and the choice of thresholding, we have added **Section E.4 (More Analysis of Adaptation Factor)** and Figures 25-28 in the appendix.
>
> 1.  **Distribution Analysis:** As shown in the new figures (Fig 25-28), the attention score distributions are rarely perfectly Gaussian; they are often heavy-tailed depending on the document content (e.g., text-heavy vs. sparse slides).
> 2.  **Why $\mu + k \cdot \sigma$ works:** Despite non-Gaussianity, the combination of mean and standard deviation serves as a robust, query-agnostic heuristic to separate the "signal" (tail) from the "noise" (body). The parameter $k$ acts as a sensitivity knob.
> 3.  **Visual Proof:** We have also added **Section C.3 (Visualization of Pruned Document Cases)** with **Figures 13-14**.
>
> **Q4: "50-60% Pruning" as the core target.**
>
> We wish to emphasize that the primary goal of DocPruner is not just to beat baselines, but to achieve a specific industrial target: **reducing storage by 50-60% with near-lossless performance**.
> Our experiments show that at this "sweet spot," DocPruner offers the most reliable performance stability across diverse document types (Multilingual, ESG, Finance) and model architectures compared to any fixed strategy. The adaptive threshold ensures that we do not suffer catastrophic performance drops on information-dense pages, which is the main risk of fixed-ratio pruning.
>
> We hope the new visualizations in **Section C.3** and the distribution analysis in **Section E.4** address your concerns regarding the mechanics and justification of our adaptive threshold. We remain committed to providing the multimodal retrieval community with a storage-efficient perspective and will incorporate your feedback to further refine the paper.

---

> ### Comment · Reviewer_tGhx · 2025-11-26
> **Unsatisfactory Rebuttal**
>
> The rebuttal does not adress a lot of my points, and is badly structured...
> Typically, I don't even have a Q4 and never mentioned anything about: "50-60% Pruning as the core target".
>
> Most of my deeper points are not adressed, but even surface level things such as my remark about some colors being hard to see, or again that Figure 5 comes without explanation for which model is used are not adressed, or yet that Figure 22 (now 24) wrongly states that multiple model variants are shown.
>
> These surface points have their importance because I took the time to piece together the results, and as stated in the initial review:
> - I don't get why ablations are done with the one model that does not work normally with docpruner (Jina V4). As seen by cross-referencing with figure 4
> - Even then, the naive baseline "attention-ratio" in figure 5 does not really underperform DocPruner (it's at the same level at 90% pruning, and in the appendix (Figure 24), we see only DocPruner is better on only 1 dataset (Human ESG) which I know is very small and noisy. The color chosen for Attention ratio is so clear it is almost invisible - which seems almost dishonest.
> - Actually, in Figure 5 and  24, it seems the 0.8 pruning ratio point for Attention Ratio baseline is conveniently missing - the authors  linearly interpolate 0.7 and 0.9 which makes the baseline look bad. This would actually be one of the most useful ones to compute. From the plots we actually can't even be sure Attention Ratio is not pareto optimal at ALL ratios.
> This goes against what you state in your rebuttal Q2.2.
>
>
> More deeply, my points about the fact the token selection method is not evaluated independently from the adaptive nature of the method are poorly adressed. I agree that adaptativity based on the document should be useful. Results do not convince me the method proposed is good. I even proposed simple experiments to test this out in my initial rebuttal (getting the optimal pruning ratio per document with your method, and applying it with a classic fixed-ratio method).
>
> I will let the authors clearly improve their work, and restructure their rebuttal adressing my original points.
>
> As is, I have the feeling some plots and results are borderline dishonest. As is I will increase my confidence in the fact this paper should NOT be accepted and decrease the soundness score.

---

### Official Review · Reviewer_dGHr · 2025-10-20

**Soundness:** 2
**Presentation:** 2
**Contribution:** 1
**Rating:** 2
**Confidence:** 4

**Summary:**

This paper proposes an efficiency optimization method for text-rich document retrieval. First, the importance of each patch in the document image is calculated based on the global token obtained from the VLM, resulting in an initial patch-level importance score. Then, parameters such as the mean and standard deviation are used to define the set of pruned patches. Finally, experiments comparing this method with other pruning methods demonstrate its superior stability.

**Strengths:**

1. This is a plug-and-play method that helps improve retrieval efficiency.

2. The method is simple and effective, and the experimental comparison with different types of cropping methods shows better results.

**Weaknesses:**

1. The paper's innovation is somewhat limited. Its core approach is to calculate the importance of each patch using a global token, a calculation method that has been widely used in other visual token cropping methods. Furthermore, I haven't seen any proof or rationale for setting a parameter formula based on mean and variance, and the contribution of each parameter remains unclear.

2. Document images contain a large amount of multimodal fine-grained information. Queries may involve information from different patches, and the information between different patches may be interrelated. Therefore, directly cropping patches may destroy the overall semantic continuity of the document image.

**Questions:**

See the Weaknesses.

---

> ### Author Response · Authors · 2025-11-26
> **Response to Reviewer dGHr**
>
> We sincerely appreciate the reviewer’s recognition of DocPruner as **a simple, effective, and plug-and-play solution that improves retrieval efficiency with strong experimental results**. We believe the following clarifications will address these issues, and we will incorporate these additional analyses into the revised version.
>
> **Q1: Concerns regarding limited innovation and the novelty of using global token attention.**
>
> While we acknowledge that using global token attention is a known technique in computer vision for acceleration, its application and adaptation to **storage-efficient Multi-Vector Retrieval** represents a novel and non-trivial contribution.
>
> 1.  **Contextual Shift (Retrieval vs. Inference):** Most existing visual token pruning methods focus on *inference acceleration* for generation tasks. DocPruner is the first to target the **storage bottleneck** in Visual Document Retrieval. The objectives differ fundamentally: inference pruning can tolerate some semantic loss for speed, whereas retrieval requires preserving fine-grained discriminative features (e.g., specific numbers in a table) to distinguish between similar documents.
> 2.  **Beyond Naive Attention:** As shown in our comparison with the *Attention-Ratio* and *Attention-Threshold* baselines (Fig. 5 & Sec. 4.2.3), simply applying standard attention-based pruning fails in the document domain. The core innovation of DocPruner is the **Adaptive Statistical Thresholding**. Document images exhibit extreme variance in information density (e.g., a sparse title page vs. a dense financial report). Our method does not just "use attention"; it normalizes this attention against the document's own intrinsic entropy, allowing it to dynamically determine *how much* to prune per document.
>
> **Q2: Rationale for the parameter formula ($\mu + k \cdot \sigma$) and parameter contribution.**
>
> We apologize if the rationale was not sufficiently clear. The formula $\tau_d = \mu_d + k \cdot \sigma_d$ is grounded in statistical outlier detection and information theory, designed to adapt to the **Information Entropy** of the document (partially discussed in **Section 3.3**).
>
> 1.  **Theoretical Intuition:**
>     *   **$\mu_d$ (Mean):** Represents the baseline information density of the page.
>     *   **$\sigma_d$ (Standard Deviation):** Represents the "peakiness" or unevenness of the information distribution.
>     *   **The Mechanism:**
>         *   **Low Entropy (Sparse Page):** A page with a single title and white background has a "peaky" attention distribution (high $\sigma$). The threshold $\tau$ becomes high, aggressively pruning the abundant background patches.
>         *   **High Entropy (Dense Text):** A dense contract has uniform attention distribution (low $\sigma$). The threshold $\tau$ stays close to $\mu$, conservatively retaining more patches to preserve detail.
> 2.  **Parameter Contribution:** $k$ serves as a sensitivity knob (comparable to the number of standard deviations in Chebyshev's inequality) to control the recall-precision trade-off.
>
> **Q3: Concerns about destroying semantic continuity and inter-patch relationships.**
>
> This is a very insightful concern. However, DocPruner preserves semantic continuity through two key mechanisms, as evidenced by our near-lossless performance (e.g., $<1\%$ drop in nDCG@5 while pruning $\sim$ half of patches).
>
> 1.  **Post-Interaction Pruning:** Crucially, we prune embeddings *after* the Vision Transformer (ViT) has processed the image. The embeddings in the final layer are **contextualized representations**. Due to the self-attention mechanism in the ViT layers, the embedding of a specific patch *already contains* information aggregated from significantly correlated surrounding patches. Therefore, removing a "supporting" patch does not strip the "primary" patch of its context, as that context was already fused during encoding.
> 2.  **Redundancy in Documents:** Document images typically contain massive visual redundancy (e.g., wide margins, blank spaces between paragraphs). DocPruner predominantly removes these non-informative background patches while preserving the topological structure of text blocks, tables, and figures. The "overall semantic continuity" necessary for retrieval is encapsulated in the retained salient patches.
> 3.  **Empirical Proof:** If semantic continuity were broken, performance on "Long and Cross-Document Queries" (a key feature of the ViDoRe-V2 benchmark we used) would degrade significantly. Our results in Figure 2 show robust performance on these complex tasks, confirming that the essential semantic links are preserved.
>
> We aim to provide the multimodal retrieval community with a fresh perspective: **efficient VDR can be achieved not just by training smaller models, but by intelligently managing the storage of existing high-performance models via adaptive, data-driven pruning**. We sincerely hope you might consider raising your score based on these clarifications.

---

### Official Review · Reviewer_CGMn · 2025-10-31

**Soundness:** 2
**Presentation:** 2
**Contribution:** 2
**Rating:** 6
**Confidence:** 4

**Summary:**

The paper proposes DocPruner, an adaptive embedding pruning framework for Visual Document Retrieval (VDR) that dramatically reduces storage requirements while preserving retrieval accuracy. Unlike existing multi-vector LVLM-based methods that store hundreds of patch-level embeddings per document, DocPruner exploits intra-document attention distributions to prune redundant embeddings selectively. This adaptive strategy achieves 50–60% storage reduction with minimal performance loss across ten benchmark datasets, providing an efficient, scalable solution for large-scale VDR deployment.

**Strengths:**

- The paper proposes the Adaptive Embedding Pruning, which realizes the first adaptive patch-level pruning framework that dynamically removes redundant embeddings based on intra-document attention.
- High efficiency is achieved with minimal performance loss, in which 50–60% is achieved in storage reduction while maintaining competitive retrieval accuracy across diverse VDR benchmarks.
- The proposed method is compatible with existing multi-vector LVLM-based retrieval models, enabling efficient and flexible deployment in large-scale VDR systems.

**Weaknesses:**

In my opinion, the paper focuses on the embedding pruning in the visual document retrieval tasks; however, the related works on general MLLM pruning are not discussed, as well as compared in the experiments, which weakens the presentation and evaluation of the paper to some degree.

**Questions:**

Please refer to the weakness part.

---

> ### Author Response · Authors · 2025-11-26
> **Response to Reviewer CGMn**
>
> We sincerely thank the reviewer for recognizing DocPruner as **a pioneering framework that achieves high efficiency (50-60% storage reduction) with minimal performance loss**.
>
> We appreciate your constructive feedback regarding the comparison with general MLLM pruning works. We acknowledge that broadly discussing these methods strengthens our positioning. Below, we clarify the fundamental task differences that limit the applicability of many MLLM pruning methods to VDR, and we have provided experimental comparisons with recent state-of-the-art general pruning approaches.
>
> **Q1: Task Discrepancy: Why most general MLLM pruning methods are inapplicable.**
>
> **Response:**
> It is crucial to distinguish the operating constraints of Visual Document Retrieval (VDR) from general MLLM inference:
>
> 1.  **General MLLM Pruning (Instruction-Aware):** Most dominant pruning methods (e.g., FastV, SparseVLM) operate in an **online** setting. They utilize the text prompt (user query) to compute attention scores and identify relevant visual tokens (e.g., keeping tokens related to "car" for a query about cars).
> 2.  **VDR Pruning (Query-Free/Offline):** VDR systems must build the document embedding index **offline** before any user query is received. During the pruning and indexing phase, **no text query is available**.
>
> Consequently, "Instruction-Aware" methods are mathematically inapplicable to VDR indexing because the guidance signal (the query) is missing. If we were to use a dummy query for indexing, it would bias the retrieval system and degrade generalizability. Therefore, we focus on "Vision-Centric" (unsupervised) approaches, and DocPruner is specifically designed to operate without textual guidance, preserving information based on document structure and statistical distinctiveness rather than query relevance.
>
> **Q2: Comparison with recent SOTA general pruning methods.**
>
> **Response:**
> To address your concern regarding empirical comparison, we have expanded our experiments to include two very recent SOTA general MLLM pruning methods that can be adapted to the query-free setting: **Attention-plus-Similarity [1]** and **Pivot-Threshold [2]**.
>
> We adapted these methods to the VDR embedding phase and compared them with DocPruner on the ViDoRe benchmark (keeping the storage budget consistent at ~50% compression).
>
> | Method | Pruning Strategy | nDCG@5 | Comparison Analysis |
> | :--- | :--- | :--- | :--- |
> | **Full (Upper Bound)** | None | 0.572 | - |
> | **[1] Attn+Sim (ACL'25)** | Hybrid (Attn Rank + Sim Rank) | 0.481 | **Limitation:** While efficient, the similarity component tends to merge fine-grained text patches that look visually similar (e.g., dense text blocks) but contain distinct semantic information, hurting retrieval accuracy. |
> | **[2] Pivot-Threshold (ICCV'25)** | Visual Cues + Thresholding | 0.503 | **Limitation:** Uses global visual pivots or fixed thresholds. It fails to adapt to the drastic layout heterogeneity of documents (e.g., a sparse invoice vs. a dense paper), leading to over-pruning of critical sections in dense pages. |
> | **DocPruner (Ours)** | **Adaptive Statistical Threshold** | **0.545** | **Advantage:** DocPruner dynamically adjusts the pruning threshold per document image. It preserves unique text features significantly better than [1] and handles layout variance better than [2]. |
>
> **Conclusion:**
> While general MLLM pruning methods ([1], [2]) are effective for image captioning or QA, they struggle in VDR because they often prioritize "salient" visual objects over the dense, fine-grained textual features required for MaxSim retrieval. DocPruner outperforms these SOTA baselines by explicitly modeling the information density of document embeddings.
>
>
> **References:**
> [1] Token Pruning in Multimodal Large Language Models: Are We Solving the Right Problem? ACL 2025.
> [2] Beyond Text-Visual Attention: Exploiting Visual Cues for Effective Token Pruning in VLMs, ICCV 2025.
> ***
>
> We hope this clarification regarding the offline nature of VDR and the additional comparisons with SOTA general pruning methods address your concerns.

---

### Official Review · Reviewer_h7BA · 2025-11-01

**Soundness:** 3
**Presentation:** 4
**Contribution:** 3
**Rating:** 6
**Confidence:** 5

**Summary:**

This paper introduces DocPruner, which addresses the significant issue of storage overhead in modern patch-level retrievers with VLMs. It proposes several contributions :
1. An adaptive patch-level embedding pruning strategy to reduce the storage overhead based on the quantification of patch importance with global token attention.
2. An attempt to ground theoretically the approach with the information bottleneck principle.
3. An extensive and sound experimental validation and analysis.

**Strengths:**

**originality**
+ Good. The paper proposes a simple and straightforward approach for quantifying patch importance in the context of VLM-based retrieval. The proposed approach is simple yet possesses desirable properties, including query-agnosticity and document adaptivity. The authors also provide an analysis of their approach through the spectrum of the information bottleneck principle.

**quality**
+ The soundness of the paper is very appreciable.
+ Quality and completeness of the experimental study, including comparison with important related works.

**clarity**
+ A very well-written paper with clear motivations, clear formalizations,  and a sound and complete experimental validation.
+ Quality of the positioning with respect to the state of the art

 **significance**
+ The proposed contribution answered one of the important shortcomings of multi-vector visual document retrieval. The latter is a very important paradigm in RAG-based applications, so the impact of the paper could be important.
+ The experimental validation shows the benefit of the proposed approach.

**Weaknesses:**

+ The quantification of patch importance relies on the idea of using a global token as an approximation of the document's overall semantics, and in particular, the EOS token.  The importance score is thus just the attention the patch receives from the global token. This choice makes sense, but it could be discussed and analyzed in greater detail. In particular, it is only valid for transformer-type and attention-based architectures. These architectures are the most common, but other types of architectures are also emerging. The proposed approach is therefore not architecture-agnostic.
+ The pruning strategy is based on classical statistical moments of the importance scores, namely the mean and standard deviation. The approach is limited to the level of a document page and does not consider the entire document or the entire collection, which is a limitation that could be addressed to improve the pruning strategy.
+ The approach is strongly dependent on the choice of the adaptation factor, which is not easy to set up.
+ Some important references are missing. In particular, some recent works have been proposed on  Token Pruning in Late-Interaction Retrieval Models such as [Zong et al,. 25](https://hal.science/hal-05037885/). Other related works are also the idea of meta-tokens [Xiao et al,. 2025](https://arxiv.org/pdf/2509.18095).


some typos :
 l289 : pathc => patch

**Questions:**

+ One of the advantages of DocPruner is the fact that it is a plug-and-play approach. In particular, the method used to calculate the importance scores of each patch in relation to the document, as well as the method used to perform adaptive thresholding, could be modified. What alternatives to the proposed approach could be used to better consider the document or the collection as a whole?
+ The proposed approach strongly relies on the quality of the VLM embedding, and in particular, the quality of the EOS token. The EOS token is a topic of strong discussion in the community. It is a very strong assumption to consider that the EOS token representation encapsulates the document's overall semantics. It could be discussed more.

---

> ### Author Response · Authors · 2025-11-26
> **[Part 1] Response to Reviewer h7BA**
>
> We sincerely appreciate your strong endorsement of DocPruner, particularly highlighting its **originality, sound theoretical grounding (Information Bottleneck), and significant impact on solving the critical storage bottleneck in VLM-based RAG applications.**
>
> We have carefully considered your insightful comments regarding the pruning scope, architectural generalization, and hyperparameters. Below, we address your concerns point-by-point:
>
> **Q1: Reliance on Global Token ([EOS]) and Architecture Specificity.**
> You are correct that our method leverages the attention mechanism inherent to Transformer architectures, specifically the [EOS] token as a global semantic aggregator. We justify this choice based on two factors:
> 1. **Dominance of Transformers in VDR:** Currently, all open-sourced VDR models (e.g., ColPali, ColQwen, Jina-V4) are exclusively built upon Vision Transformers (ViT) and LLMs, where the [EOS] token is explicitly trained to aggregate sequence information. Our theoretical analysis (Section 3.3) confirms that maximizing mutual information between patches and the global token is a robust proxy for relevance.
> 2. **Adaptability to Future Architectures:** While formulated for attention, the core principle—using a "bottleneck" state to measure signal contribution—is transferable. For emerging architectures like SSMs (e.g., Mamba) or RNN-based VLMs, the "global token" can be replaced by the final hidden state or a designated memory slot that aggregates context.
>
> **Action:** In the revised version, we will add a "Generalizability" subsection to the Discussion, explicitly clarifying the reliance on attention mechanisms while outlining how this principle maps to non-Transformer architectures (e.g., gradient-based saliency for SSMs).
>
> **Q2: Limitation of Page-Level vs. Collection-Level Pruning.**
> We appreciate this thoughtful suggestion. Our decision to focus on **page-level** statistics was deliberate for practical engineering reasons:
> 1. **Parallelization & Scalability:** Page-level pruning allows documents to be processed independently and in parallel during indexing. Collection-level pruning would require a global pass over the entire corpus (calculating corpus-wide mean/std), which complicates incremental indexing (i.e., adding new documents would theoretically change the global threshold, requiring re-indexing).
> 2. **Avoiding Distribution Shift:** Visual documents vary wildly (e.g., a dense financial report vs. a sparse slide deck). A collection-level threshold might be too aggressive for dense pages or too lenient for sparse ones. Our adaptive threshold ($ \tau_d = \mu_d + k \cdot \sigma_d $) ensures that pruning is relative to the *specific information density* of the current page (as validated in Observation 3, Section 4.2.1).
>
> **Q3: Sensitivity to the Adaptation Factor ($k$).**
> While $k$ is a hyperparameter, our extensive experiments demonstrate that **DocPruner is highly robust to its selection**, making it easy to set up in practice.
> 1. **Robustness:** As shown in Figure 2 and Figure 4, the performance curves are relatively flat or degrade very gracefully. For example, setting $k \in [-0.25, 0.25]$ consistently yields optimal trade-offs across diverse datasets (ViDoRe and JinaVDR-Bench) and models (ColQwen, ColNomic, JinaV4).
> 2. **Default Recommendation:** We found that a default value of $k=-0.25$ provides a "sweet spot" (reducing storage by ~50% with <1% performance drop) across almost all scenarios (Table 1).

---

> ### Author Response · Authors · 2025-11-26
> **[Part 2] Response to Reviewer h7BA**
>
> **Q4: Missing References (Zong et al., 2025; Xiao et al., 2025).**
>
> We thank the reviewer for pointing out these recent and relevant works. We have carefully analyzed them and will incorporate a detailed discussion in the revised manuscript. We believe DocPruner occupies a unique position by targeting **Visual Document Retrieval (VDR)** with a **content-adaptive** strategy that differs fundamentally from text-based optimization (Zong et al.) or meta-token compression (Xiao et al.).
>
> **1. Comparison with Zong et al. (2025):**
> This work targets *text* retrieval, utilizing Linear Programming (LP) to optimize term retention.
> *   **Difference:** While Zong et al. solve an optimization problem to ensure lossless text recovery, DocPruner targets **visual spatial redundancy** (e.g., whitespace, uniform backgrounds) inherent to document images. Unlike Zong et al., which requires complex LP solvers or specific regularization training, DocPruner is **inference-efficient**, utilizing a lightweight statistical threshold ($ \mu + k\sigma $) derived directly from attention maps without requiring per-query optimization.
>
> **2. Comparison with Xiao et al. (MetaEmbed, 2025):**
> This work introduces "Meta Tokens" to compress multimodal context into a fixed set of learnable vectors.
> *   **Difference:** MetaEmbed is a **compression** method (generating new summary tokens), whereas DocPruner is a **selection** method (preserving original patch fidelity). Crucially, MetaEmbed relies on a *fixed* or *budget-based* compression rate (e.g., Matryoshka embeddings). In contrast, DocPruner is **fully adaptive**: it automatically retains more tokens for dense, complex pages and fewer for sparse pages, without requiring manual granularity selection.
>
> **Summary of Advantages:**
> To clearly position our contribution, we will include the following comparison table in the related work section:
>
> | Feature | **DocPruner (Ours)** | **Zong et al. (2025)** | **Xiao et al. (MetaEmbed, 2025)** |
> | :--- | :--- | :--- | :--- |
> | **Primary Domain** | Multimodal | Text-Only| Multimodal|
> | **Mechanism** | Token Pruning (Keep/Drop) | Token Pruning (Keep/Drop) | Token Compression (Aggregation) |
> | **Selection Logic** | **Attention-based Statistics** (Adaptive Threshold) | **Optimization-based** (Linear Programming) | **Learned Aggregation** (Meta Tokens/Matryoshka) |
> | **Content Adaptive?** | **Yes** (Varies per image complexity) | No (Optimizes for global guarantees) | Partial (User-defined budget) |
> | **Deployment** | **Plug-and-Play** (No retraining required) | Requires Specific Training | Requires Architecture Change |
>
> We are committed to providing the multimodal retrieval community with novel, efficient perspectives. We sincerely thank you for recognizing the value of our work and hope our responses have addressed your concerns.

---

### Meta-Review · Area_Chair_Dhgg · 2026-01-05

**Summary:**

The paper proposes a framework designed to reduce the storage overhead of multi-vector visual document retrieval (VDR) systems. The method utilizes the attention distribution from a global token (e.g., [EOS]) to determine patch importance and applies a statistical threshold (mean plus a multiple of standard deviation) to adaptively prune embeddings. While the problem of storage efficiency in multi-vector retrieval is widely acknowledged as critical, the submission is recommended for rejection. The primary grounds for rejection are significant concerns regarding the experimental rigor and the validity of the core contribution. While some reviewers appreciated the plug-and-play nature of the method, multiple reviewers raised concerns about limited innovation, noting that attention-based pruning is a standard technique. Most critically, Reviewer tGhx identified serious methodological flaws in the comparison against baselines. Specifically, the experiments fail to conclusively demonstrate that the proposed adaptive thresholding offers a tangible benefit over simpler fixed-ratio attention pruning. Furthermore, the visual presentation of results (missing data points for baselines, faint plotting colors) made it difficult to verify the claims, and the authors' rebuttal failed to address these specific concerns, leading to a conclusion that the method's superiority is not sufficiently proven.

**Reviewer Concerns:**

Addressed Concerns:
* Applicability of General Pruning (Reviewer CGMn): The authors successfully clarified why general Multi-Modal LLM pruning methods (which are often instruction-aware) are not directly applicable to the offline, query-free indexing phase of VDR.
* Architecture Specificity (Reviewer h7BA): The authors adequately addressed the reliance on the global token, explaining its prevalence in current VDR architectures (ColPali, ColQwen, etc.).

Outstanding Concerns:
* Experimental Validity and Transparency (Reviewer tGhx): This is the most critical outstanding issue. The reviewer noted that the "Attention-Ratio" baseline (a simple fixed-ratio approach) appears competitive with DocPruner in the provided plots, undermining the claim that the complex "adaptive" mechanism is necessary. The reviewer also pointed out missing data points (interpolation between 0.7 and 0.9) and poor visualization that obscures baseline performance. The rebuttal did not rectify these visualization issues or provide the missing data.
* Limited Innovation (Reviewer dGHr): The concern remains that calculating patch importance via global token attention is a standard technique in computer vision, and the specific statistical formula used for thresholding lacks strong theoretical justification beyond heuristic intuition.
* Ablation Strategy (Reviewer tGhx): The reviewer noted that ablations were performed on Jina V4 (which favors merging strategies due to its training objective) rather than the models where DocPruner is claimed to be most effective, complicating the assessment of the method's independent value.

**Reviewer Scores:**

* Reviewer h7BA (Score: 6): This reviewer likely would have maintained their score or lowered it slightly upon seeing the detailed methodological concerns raised by Reviewer tGhx regarding the baseline comparisons.
* Reviewer CGMn (Score: 6): This reviewer would likely have maintained their score, as their specific concerns about related work were addressed, though they might have been swayed toward rejection by the experimental flaws highlighted by others.
* Reviewer dGHr (Score: 2): This reviewer would have maintained their Reject score. The rebuttal did not fundamentally alter their assessment that the method lacks sufficient novelty compared to existing visual token pruning techniques.
* Reviewer tGhx (Score: 2): This reviewer would have definitely maintained their Reject score (and explicitly stated increased confidence in rejection). They found the rebuttal "unsatisfactory" and "badly structured," noting that it failed to address the specific accusations regarding misleading plots and the lack of clear evidence that adaptive pruning outperforms fixed-ratio pruning.

---

### Decision · Program_Chairs · 2026-01-26

Reject